# Reassessing long-standing meteorological records: an example using the national hottest day in Ireland

Katherine Dooley[1,2,] Ciaran Kelly[1], Natascha Seifert[1], Therese Myslinski[1], Sophie O'Kelly[1], Rushna Siraj[1], Ciara Crosby[1], Jack Kevin Dunne[1], Kate McCauley[1], James Donoghue[1], Eoin Gaddren[1], Daniel Conway[1], Jordan Cooney[1], Niamh McCarthy[1], Eoin Cullen[1], Simon Noone[1], Conor Murphy[1], Peter Thorne[1]

[1] Irish Climate Analysis and Research UnitS (ICARUS), Maynooth University, Maynooth, Co. Kildare, Ireland.

[2] Environmental Protection Agency, PO Box 3000, Johnstown Castle Estate, County Wexford, Ireland, Y35 W821.

*Correspondence to*: Katherine Dooley (Ka.Dooley@epa.ie)

**Abstract.** This analysis highlights the potential value in reanalysing early national meteorological records from around the world. These were oftentimes measured via techniques that preceded standardisation of instrumentation and methods of observation and thus could be subject to considerable biases and uncertainties. It uses the techniques pioneered by WMO record assessment teams. The highest currently recognised air temperature (33.3°C) ever recorded in the Republic of Ireland was logged at Kilkenny Castle in 1887. The original observational record however no longer exists. Given that Ireland is now the only country in Europe to have a national heat record that was set in the 19th century, a reassessment of the verity of this record is both timely and valuable. The present analysis undertakes a fundamental reassessment of the plausibility of the 1887 temperature record using methods similar to those used to assess various weather extremes under WMO auspices over recent years. Specifically, we undertake an inter-station reassessment using sparse available records and make recourse to the new and improved 20CRv3 sparse-input reanalysis product. Neither surrounding available stations nor the reanalysis offer substantive support for the Kilkenny record of 33.3°C being correct. Moreover, recent data rescue efforts have uncovered several earlier extreme values, one of which exceeds the Kilkenny value (33.5°C on 16th July 1876 recorded at the Phoenix Park). However, the sparsity of early observational networks, a distinct lack of synoptic support from 20CRv3 for many of the extreme heat values, and the fact that these measurements were obtained using non-standard exposures leads us to conclude that there is grossly insufficient evidence to support any of these 19th Century extremes as robust national heat record

candidates. Data from the early 20th Century onwards benefits from a denser network of stations undertaking measurements in a more standardised manner, many under the direct auspices of Met Éireann and its predecessors, adhering to WMO guidance and protocols. This enables more robust cross-checking of records. We argue that the Met Éireann recognised 20th Century heat record from Boora in 1976 verifies as the most plausible robust national temperature record based upon the synoptic situation and comparisons with nearby neighbouring stations. This measurement of 32.5°C thus likely constitutes the

highest reliably recorded temperature measurement in the Republic of Ireland. Ultimately, the formal decision on any reassessment and reassignment of the national record rests with the national meteorological service, Met Éireann.

## 1. Introduction and Context

National meteorological records of heat, cold, precipitation and other meteorological parameters play a key role in the

communication of weather events and climate change at both national and international scales. Over the past decade or so the WMO has instigated teams of experts to assess or reassess global and regional records for a range of phenomena. These have either recognised (Queterlard et al., 2009, Courtney et al., 2012, Purevjav et al., 2015, Lang et al., 2017, Cerveny et al., 2017, Láska et al., 2018, Merlone et al., 2020, Peterson et al., 2020, Weidner et al., 2021) or occasionally removed and replaced (El Fadli et al., 2013) various global or regional records for meteorological phenomena including heat and cold records. The latter

revocation by El Fadli et al. was of particular note as it removed a long-standing global all-time heat record which had appeared e.g. in the Guinness book of world records, encyclopaedias and other similar reference materials. In comparison, national records have, at least in most cases, received scant attention. Yet the tools and approaches employed by the WMO are eminently scalable to a national context. Here, by use of Ireland's reported all-time heat record as an example, we make the case for assessing anew national records using the tools and techniques pioneered by these WMO expert teams. Such investigations

are particularly valuable for early records that occurred prior to standardisation of instrumentation and methods of observation when the biases in the original records may be substantial (Böhm et al. 2010, Brunet et al. 2011, Murphy et al. 2020).

The highest recognised shaded air temperature recorded in Ireland was observed at Kilkenny Castle on 26th June 1887 (Rohan,

1986). Kilkenny Castle is located at 52.6505° N, 7.2493° W, 60 metres above sea level. In 1841, the population of Kilkenny City was approximately 19,071, decreasing to 15,257 in 1851 (McDill, 2021). According to the 2016 census, Kilkenny City now has a population of 25,512 (City Population, 2021). The reported temperature was observed from a "standard site," reading 33.3°C (Rohan, 1986:133). Unfortunately, the original records from Kilkenny Castle for the period are missing and unavailable for scrutiny (Mary Curley, pers comm). Curiosity surrounds this temperature record as Ireland remains the only country in

Europe to have an existing heat record from the 19th century, with almost all other European countries having had their heat records set much more recently (Table 1). Given this, and the relative paucity of metadata supporting the Irish record, it is appropriate to re-examine this record using modern investigative techniques.

| Country | Temperature (°C) | Date | Location |
|---|---|---|---|
| Republic of Ireland | 33.3 | 26th Jun 1887 | Kilkenny Castle (Kilkenny) |
| England | 38.7 | 25th Jul 2019 | Cambridge Botanic Garden |
| Wales | 35.2 | 2nd Aug 1990 | Hawarden Bridge (Flintshire) |
| Scotland | 32.9 | 9th Aug 2003 | Greycrook (Scottish Borders) |
| Northern Ireland | 31.3 | 21st July 2021 | Castlederg (County Tyrone) |
| France | 46.0 | 28th Jun 2019 | Veragues, Herault |
| Belgium | 41.8 | 25th July 2019 | Begijnendijkm Flemish Brabant |
| Netherlands | 40.7 | 25th July 2019 | Gilze en Rijen |
| Norway | 35.6 | 20th June 1976 | Nesbyen, Buskerud |
| Iceland | 30.5 | 22nd June 1939 | Teigarhorn, Djupivogur |

**Table 1: Highest daily maximum temperature records of several NW European countries surrounding Ireland including constituent**
**countries of the United Kingdom sourced from various National Meteorological Service websites at the time of writing.**

The national temperature record was logged at Kilkenny Castle using a Negretti and Zambia thermometer, a standard thermometer in use at the time. This thermometer was tested by the Meteorological Council in 1890 which showed the instrument required no corrections, thus reporting the true value (Report of the Meteorological Council, 1890). During the

1880s the Stevenson Screen (Stevenson, 1864) became the preferred standard screen to shield thermometers from sunlight and radiation. Dolores Gaffney, the collections officer at Kilkenny Castle, believes the temperature in question was recorded within a Stevenson Screen (pers. comm.). The original Kilkenny Castle site was surveyed in 1890 but not considered a 'second order' (maintained by trained staff (UCAR, 2021)) station (RMS, 1890) and eventually closed in 1933 (Niall Dollard, pers. comm.). The exact location of the weather station in Kilkenny Castle at the time of the record is unknown. Station siting is known to

have substantial potential impacts upon the representativity of resulting measurements and modern guidance from WMO specifies criteria for siting (WMO, 2018). Kilkenny Castle is in the centre of Kilkenny Town beside the River Nore (Figure 1). Given ample grounds it is possible that the instrument was well sited but, equally, there are many potential sitings which would today be considered to yield the possibility of biased records. Met Éireann explained that a station inspection report from approximately 1911 stated that Kilkenny Castle only kept the records for four years before destroying them (Mary Curley,

pers. comm.). This both means that the original record from which the national temperature record arises is lost for perpetuity and that necessary metadata to understand the measurement context are also unavailable. Unfortunately, the timeseries of daily recordings around the record heat event were not saved and so the single 'observation' of record heat must be analysed in isolation.

This added challenge of missing data is not an isolated occurrence. The crossover of manual to digitised recordings, early inconsistencies in logs before standardisation and the destroying of records viewed as 'no longer useful' has been reported on at both European and global scales (Brunet and Jones, 2011). Data rescue efforts are being recognised as imperative tools in the understanding of early climate variability and change (Allan et al., 2011, Brönnimann et al 2019), but are of limited utility for stations that had a policy of actively disposing of their records as is the case at Kilkenny Castle.

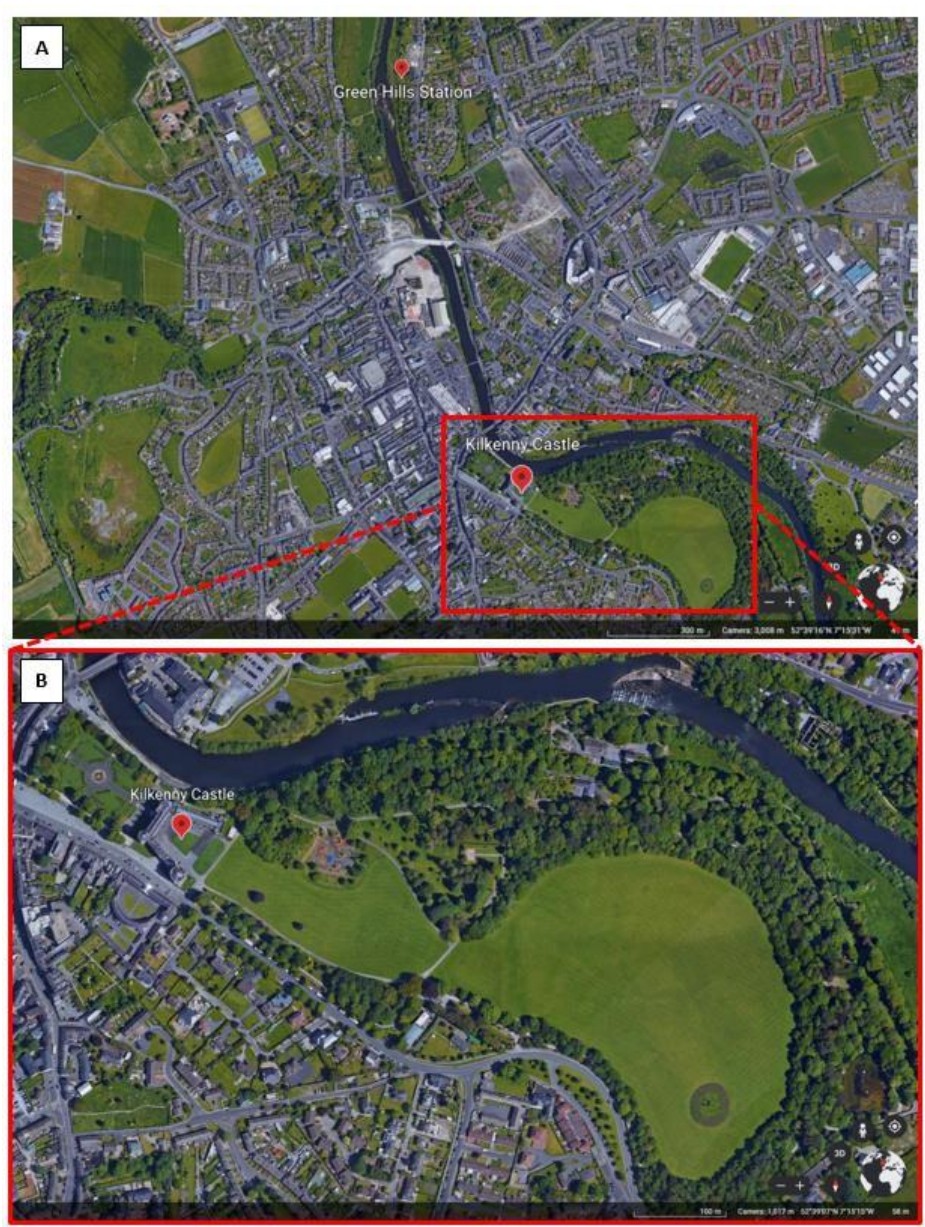


**Figure 1. Aerial Image of Kilkenny Castle and its' surrounding area (top) and the grounds of the castle (bottom) (©Google Earth v. 9.152.0.1 (2021)).**

Met Éireann also recognises a 20th Century highest air temperature of 32.5°C recorded on 29th June 1976 in Boora, County Offaly. This reading came during the long heatwave and summer drought which affected the British and Irish Isles

along with much of NW Europe (Stubbs, 1977, Noone et al., 2017). This station recorded data from 1950 to 2016 and was maintained by Met Éireann. Based upon available digital data, other notable heat events in the national instrumental record are summarised in Table 2. These include a suite of new potential heat records arising from recent highly valuable data rescue efforts as part of a collaboration between Met Éireann and National University of Ireland, Galway as part of a PhD project (Mateus et al., 2020). Notable heat extremes have occurred exclusively in meteorological high summer – late June through

early August.

| Location | Date | Longitude | Latitude | Height above sea level | Value (°C) | Source and notes |
|---|---|---|---|---|---|---|
| Phoenix Park, Dublin | 16/07/1876 | 53.37" N | 6.33° W | 48m | 33.5 | Mateus et al. (2020) |
| Kilkenny Castle, Kilkenny | 26/06/1887 | 52.65° N, | 7.25° W | 60m | 33.3 | Met Éireann's recognised national record |
| Dunmore East, Waterford | 29/06/1851 | 52.15° N | 7.00° W | 20m | 33.3 | Mateus et al. (2020) |
| Markree, Sligo | 28/06/1851 | 54.18° N | 8.46° W | 34m | 33.3 | Mateus et al. (2020) |
| Markree, Sligo | 27/06/1852 | 54.18° N | 8.46° W | 34m | 33.1 | Mateus et al. (2020) |
| Markree, Sligo | 26/06/1853 | 54.18° N | 8.46° W | 34m | 32.9 | Mateus et al. (2020) |

| | | | | | | |
|---|---|---|---|---|---|---|
| Kilrush, Clare | 28/06/1851 | 52.64° N | 9.48° W | 24m | 32.8 | Mateus et al. (2020) |
| RCS, Dublin | 03/08/1856 | 53.34° N | 6.26° W | 29m | 32.8 | Mateus et al. (2020) |
| Boora, Offlay | 29/06/1976 | 53.22° N | 7.72° W | 58m | 32.5 | Met Éireann's recognised national 20th Century record |
| Elphin, Roscommon | 19/07/2006 | 53.85° N | 8.19° W | 92m | 32.3 | Warmest July day in Met Éireann records |
| Oak Park, Carlow | 14/07/1983 | 52.86° N | 6.92° W | 58m | 32.2 | Met Éireann |
| Killarney, Kerry | 12/07/1921 | 52.07° N | 9.51° W | 55m | 32.2 | Mateus et al. (2020) |
| Markree, Sligo | 29/06/1854 | 54.18° N | 8.46° W | 34m | 32.2 | Mateus et al. (2020) |
| Scattery Island, Clare | 07/08/1851 | 52.61° N | 9.51° W | 5m | 32.2 | Mateus et al. (2020) |
| Shannon Airport, Clare | 28/06/2018 | 52.69° N | 8.92° W | 15m | 32.0 | Met Éireann |
| Dooks, Kerry | 18/07/2006 | 52.07° N | 9.93° W | 15m | 32.0 | Met Éireann |
| Ballybrittas, | 29/06/2006 | 53.06° N | 7.08° W | 90m | 32.0 | Met Éireann |

| Laois | | | | | | |
|---|---|---|---|---|---|---|
| Boora, Offlay | 02/07/1976 | 53.22° N | 7.72° W | 58m | 32.0 | Met Éireann |

**Table 2. Notable heat extremes in available records from the Republic of Ireland over the instrumental record sourced from Met Éireann's website and available digital records, including the recently rescued early holdings from Mateus et al. (2020). The table includes all records that exceed 32˚C at least one of which must plausibly constitute the robust national record.**

Given the uncertainty associated with the Irish national high temperature record, together with its novelty in being the earliest national temperature record in Europe, the 2019-20 class of the MSc Climate Change at Maynooth University were set a group assignment to re-evaluate the record. This paper represents the outcomes of that reassessment. Having introduced in the present section the national context and aspects of the recognised all-time and 20th Century national heat records, as well as selected additional heat extremes in available records, the rest of the paper is structured as follows. Section 2 introduces key methodological approaches and results from past WMO extremes verification efforts which this exercise was designed to mimic. Section 3 goes on to apply these approaches to propose that the long-standing Kilkenny Castle national heat record should be rejected. Section 4 then considers additional candidates arising from Table 2, until alighting upon a recommendation for the most plausible highest reliable heat record in the available observations. Section 5 concludes.

## 2. Assessments and Certifications of Climatic Extremes

The WMO has instigated a formal process to verify and certify a broad range of extremes. A rapporteur, when advised of a candidate record extreme that is desired to be assessed convenes a team of domain area and regional experts to investigate the event and recommend either acceptance or rejection as a bona fide observation. If the event is rejected the team is asked to recommend an alternative record if they can do so. Teams have been instigated for temperature (El Fadli et al., 2013, Láska et al., 2018, Merlone et al., 2020, Weidner et al., 202), precipitation (Quetelard et al., 2009), winds (Courtney et al., 2012) and pressure (Purevjav et al., 2015) amongst other records. The WMO teams consider global or WMO region records and have not, to date, assessed national records.

Teams assess all aspects of the potential record including available metadata, data, the synoptic situation and, if possible, the instrumentation, to come to a conclusion. The extent of the investigation depends upon the nature and timing of the event being analysed. More recent events can benefit from more thorough analyses which may include instrument characterisation and site

visits (Merlone et al, 2020). Whereas revisiting old records, as is the case herein, has to rely more on event characterisation and reference to nearby instrumental records (e.g. El Fadli et al., 2013). Increasingly, recourse is made to meteorological reanalysis products, including sparse-input centennial scale reanalyses (Slivinski et al., 2019) that can provide information on the evolving synoptic meteorological situation surrounding the record event.

Several analyses of various meteorological records worldwide have been undertaken over recent years leading to their removal (WMO, 2021). Perhaps most famously, what was cited to be the world's highest recorded temperature for over 90 years was invalidated following a careful WMO-team reappraisal. A temperature of 58°C was recorded at El Azizia (modern day Libya) but was disproved due to various issues brought to light after an in-depth investigation of the record. The issues surrounding the reading included problematic instrumentation, siting, observer interpretation and, notably in

the context of the present study, that the record did not correspond to other nearby locations (El Fadli et al., 2013). The South African Weather Service (SAWS, 2019) reported a temperature of 50.1°C from the Vioolsdrif weather station on the 28th of November 2019 (Austral summer). This reading exceeded all previous heat records for South Africa. However, the temperature sensor in the Vioolsdrif weather station was replaced two days prior to the record temperature and upon tracking the behaviour of the station over the following days, it was decided that the behaviour of the

temperature sensor was questionable. This resulted in the 50.1 °C record being deemed invalid. A temperature of 42.9 °C was recorded at the Deelen weather station, in the Netherlands, on the 25[th] of July 2019 (Schildkamp, 2019). The sensors at the Deelen station were inspected and any technical faults were ruled out. However, another instrument on the same site, a few hundred metres away, didn't record the same high values, which resulted in the record being dismissed. Lastly, a high temperature of 53°C recorded in Cloncurry, Queensland, Australia in 1889 was disregarded

due to faults in the measurement and was therefore not included in Australia's Bureau of Meteorology's list of recognised extremes (Smith, 2012).

Of those assessments of possible records, the most analogous to the present analysis is that of El Fadli et al. (2013) in that the original instrumentation has long since disappeared and the metadata is imperfect. Like El Fadli et al. (2013), we therefore must make recourse to nearby stations and sparse-input centennial timescale reanalyses to reassess the Kilkenny record.

### 3.  A Critical Reassessment of the Kilkenny Castle National Heat Record

Figure 2 displays the location of the Kilkenny station as well as the six closest stations with continuously available daily records over the period of the national heat record. There are very few stations with continuous daily records over this period, with only 5 other stations on the island of Ireland for which records had been digitised and made available at the time of the present analysis. Data was sourced from holdings under preparation for the Copernicus Climate Change Service that arise from
Met Éireann and the UK Met Office. Additional records may exist in the Met Éireann archives that are yet to be digitised or those of the UK Met Office, and the analysis largely preceded the availability of the very recently rescued holdings from Mateus et al. (2020) (Table 2). The stations on the island of Ireland all consist of long-running sites that have been well maintained either by Met Éireann and its predecessors or the Armagh Observatory. To augment the records on the island of Ireland, recourse is also made to records from Sheffield in the UK. There are other British stations for the same time but they
are even further from the Kilkenny site (e.g. Oxford, Kew Gardens). With the notable exception of a station situated south-east of Kilkenny, the stations provide a reasonable geographical spread sufficient to infer spatial temperature gradient behaviour on the day of the record.

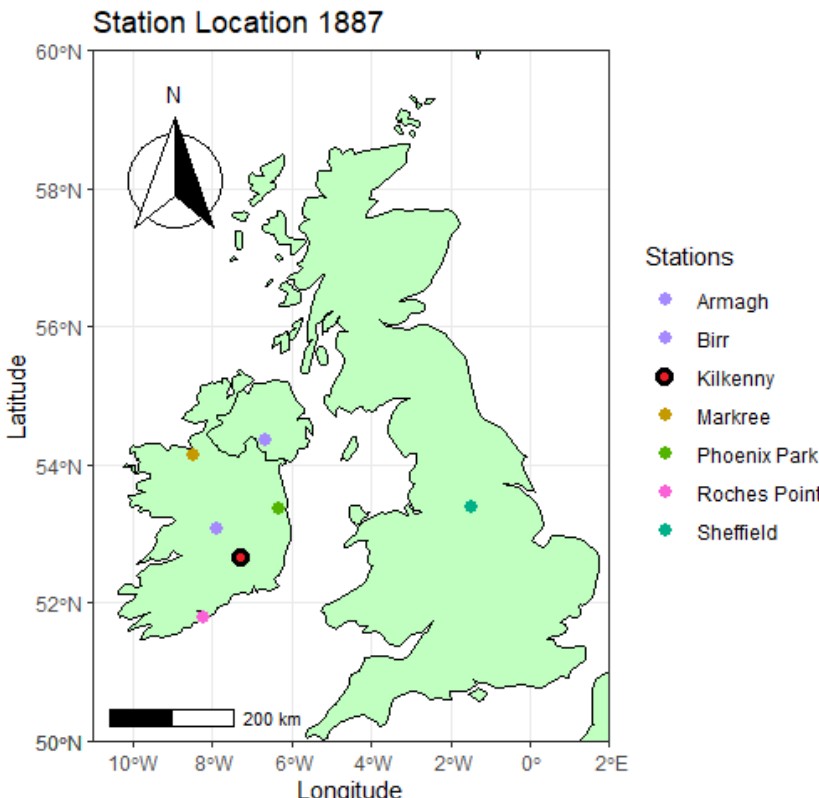

**Figure 2. Location of the seven stations in Ireland and England used in this study to assess the reliability of the reported Kilkenny record. The green marker shows the location of the Kilkenny station. Station selection is based upon availability over the period of the Kilkenny record high temperature event.**

Figure 3 illustrates observed surface temperatures for the month of June 1887 for the six comparator weather stations and the reported record temperature from Kilkenny. All station series show similar temporal evolution in their maximum daily temperatures through June of 1887 (although there is some ambiguity as to the timing of daily observations at several sites). A period of sustained heat builds during the second half of June peaking on or around the day of the record, before breaking the next day. The break in heat appears to progress from north to south. Geographically, the

nearest recording to the Kilkenny record on the 26[th] was Birr, which reported 29.3°C, followed by Roches Point which reached 22.2°C. These high temperatures are backed up by contemporary records of the summer of 1887 being hot and dry. Noone et al. (2017) note that 1887 is one of the most intense drought years in Ireland in precipitation records spanning the past 250 years. Barrington (1888) assessed the impacts of the previous year's drought on agriculture, describing the drought of 1887 as being most extreme in the south and southeast, particularly in Counties Kilkenny, Wexford and Cork where April to June precipitation was as little as 30 percent of normal. Barrington also notes widespread crop failure across Ireland in 1887, while newspaper articles from the time indicate reduced harvests and crop failure throughout the country. Industrial activity, particularly the Linen industry in Northern Ireland was also adversely affected (Noone et al., 2017). Although the main focus of Barrington (1888) is on precipitation deficits, he does also note the temperature "*On Sunday, June 5th, the temperature rose, and for the remainder of the month we had a combination of heat and drought, which lasted until July 10th. No record exists of such a hot and dry June in the south and southeast of Ireland*". Barrington (1888) goes on to mention that *"... in June and July 1887, the S. of Ireland suffered more from excess of heat than any part of England or Scotland."* further cementing the plausibility that the Island of Ireland was experiencing extreme hot and dry conditions at the time of the Kilkenny record high.

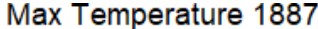

**Figure 3. The maximum air surface temperatures of the six selected weather stations for the month of June 1887. The red point marker displays the maximum surface temperature recorded at Kilkenny on the 26th June 1887.**

The 20CRv3 reanalysis product (Slivinski et al., 2019) extends back to the early 19th Century using surface pressure

185    and prescribed sea surface temperature boundary conditions to create physically consistent reconstructions of climate.

This research paper gathered analyses at 1500GMT (1430 LST) – the closest 3-hourly analysis window to the expected

timing of daily maxima in Ireland - using the ensemble mean product. Examination of 20CRv3 fields over the period

around 26th June 1887 highlights that there was a high-pressure system in place throughout the examined period (Figure

4). Prior to the 26th June the high pressure is centred to the north of Ireland advecting air from the near continent. On 26th

190 June the high pressure collapses and is subsequently replaced by a building high pressure from the near Atlantic initially advecting cooler air from the north which becomes centred over Ireland by the end of the month. The evolving pressure situation is highly consistent with the timing and relative phasing of changes reported from each site in Figure 3.

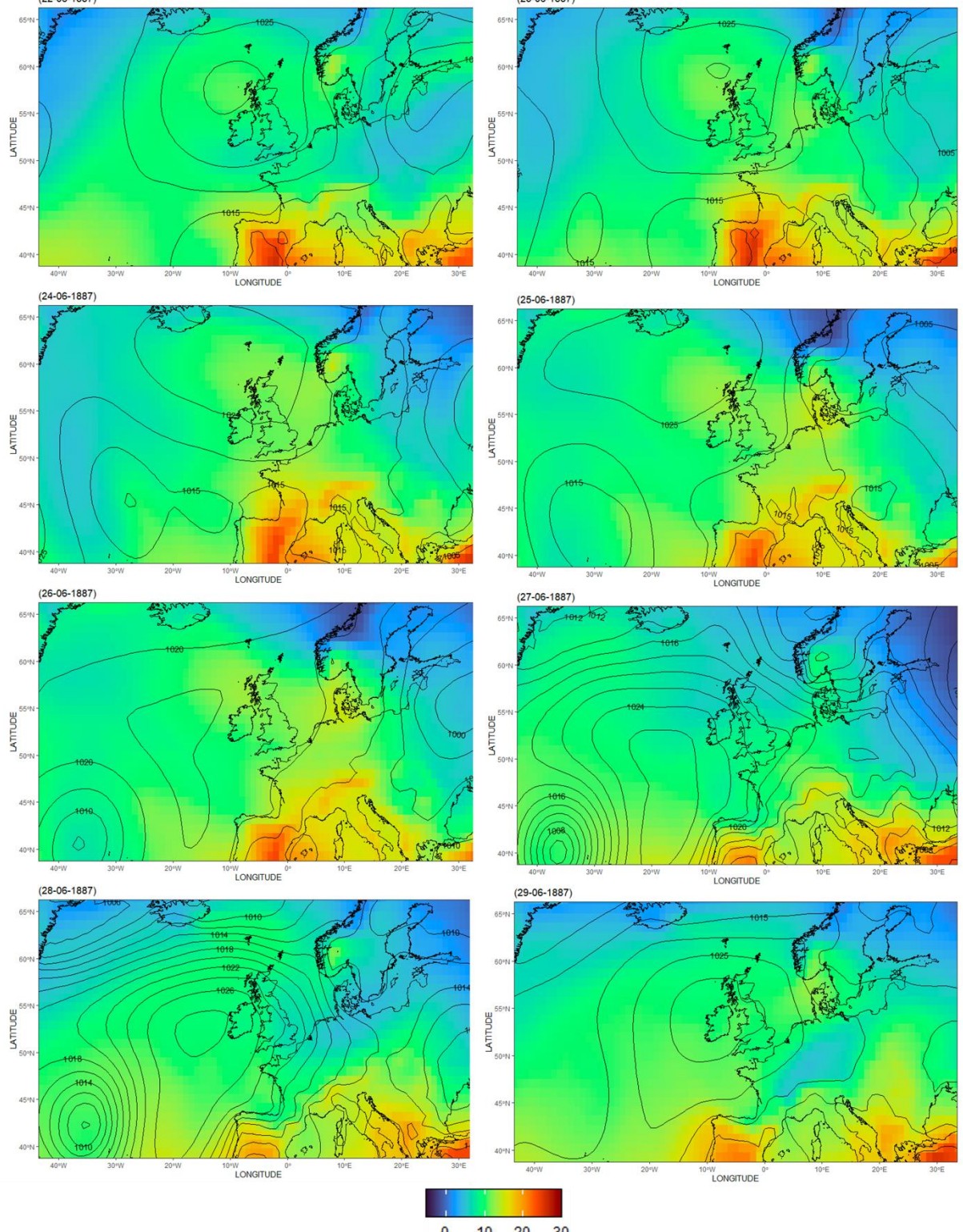

**Figure 4. NOAA 20CRv3 MSLP and 850-hPa Temperatures over an 8 day period centred upon the 26th June 1887. 20th Century Reanalysis V3 data provided by the NOAA/OAR/ESRL PSL, Boulder, Colorado, USA, from their Web site at https://psl.noaa.gov/data/gridded/data.20thC_ReanV3.html. The domain of latitude 35N-70N and longitude 45W-30E was chosen to provide a synoptic view, with Ireland centred to provide an overview of the surrounding regions. 850hPa temperatures are given in colours while MSLP is given as contours. All maps correspond to 1500 UTC which is the approximate expected time of daily maxima in Ireland (corresponding to c. 14.30 LST).**

On the basis of the synoptic situation (Figure 4) and the contiguous station series (Figure 3) it is certain that the period was indeed a notable heatwave event that could, plausibly, be consistent with a record high temperature. From the 22nd – 26th Ireland lay in a slack south east to east airflow with an anticyclone of 1025 hPa centred to the north of the United Kingdom and Ireland and a weak cut-off cyclone of 1015 hPa centred to the west of Iberia. This situation would be typical for the advection of continental tropical airmasses and the slack gradient consistent with low winds enabling local diurnal solar heating of the near-surface airmass. Taken together, these conditions would be broadly consistent with exceptional warmth relative to seasonal expectations. However, such a synoptic situation is not uncommon in the summer season over Ireland. In isolation this evidence is thus insufficient to determine the validity of the record. On the 27th-28th a new anticyclone of 1024 hPa replaced the previous anticyclone and was centred just to the west of Ireland. There was a deepening cyclone almost stationary to the west of the Canary Islands and a developing cyclone centred just to the west of Norway. This resulted in a northerly / north westerly flow of advection of polar maritime air over the 26th-27th leading to the cooling trend seen at all contiguous station series (Figure 3). By the 29th the anticyclone became centred over Ireland with slack winds. The evolving synoptic situation from 20CRv3 is entirely consistent with the temporal evolution of temperatures at the stations plotted in Figure 3.

The 850hPa temperature, which corresponds approximately the top of the boundary layer in Ireland in summer, is often used by meteorologists to indicate the potential surface temperatures and is used here and elsewhere to indicate the potential for significant surface warmth. However, under high pressure and stable air masses there is considerable potential for near surface inversions that enable build-up of additional heat near the surface above and beyond what would be expected from simple adiabatic lapse rate assumptions. The relative coarse resolution of the 20CRv3 product (1 degree by 1 degree) and potential for climatological biases in surface temperatures arising from this arguably make the 850hPa temperatures more reliable for the present analysis than the 2m air temperature estimates as an indicator of at least the potential for significant and widespread

surface warmth. Focussing in more locally and considering the 850hPa reanalysed temperature fields (Figure 5) highlights an increase in temperatures at this pressure level, peaking around the 25$^{th}$, followed by a rapid cooling and then a degree of recovery through the period. The 850hPa temperatures are not, however, exceptional, reaching only the mid- to high-teens at their peak. While this cannot preclude very high surface temperatures, it would require a very substantial near surface inversion

to support temperatures at the surface as high as that reported from Kilkenny Castle.

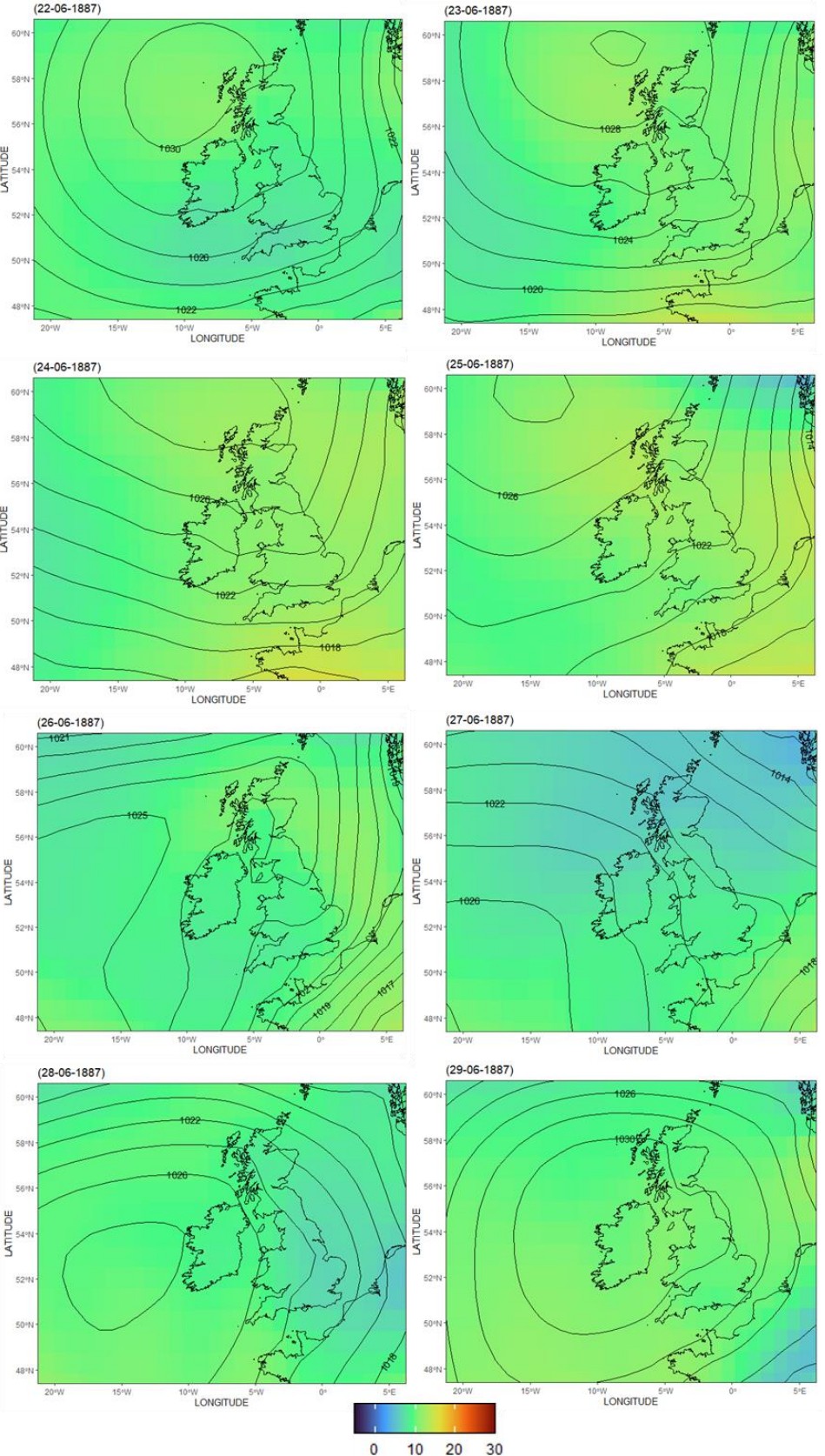

**Figure 5. As Figure 4 but zoomed in to the British and Irish isles region. See Figure 4 caption for further details and note that 20CRv3 native resolution is 1 degree by 1 degree.**


With the exception of the Kilkenny Castle site, all sites used in Figure 3 have nearly complete records through to present, many of which have been digitised. Since 2010, data from Kilkenny Greenshill (also referred to as Green's Hill or Green Hills), which is very close to the location of the original Kilkenny Castle site, is available via Met Éireann. The Greenshill site is at a distance of 1.6 km NNW from Kilkenny castle (Figure 6). The elevation differs by approximately 12 m with Kilkenny castle

lying at 58m asl while the Greenhills station lies 46m asl. Both sites are reasonably proximal to the River Nore, and both are surrounded by some urban infrastructure. Thus, the Kilkenny Castle and Greenshill stations are likely to have been broadly comparable. Although clearly the lack of a period of overlap leaves some irreducible uncertainty in this interpretation. The availability of several years of modern data using modern instrumental techniques both near the site of the original Kilkenny record and at the sites of the historical comparator sites permits statistically based comparisons. Summer season timeseries

from each station were matched to account for missing data and then modern differences between station pairs analysed to provide context for the evident offsets between Kilkenny Castle and remaining sites in Figure 3 on 26th June 1887. For Birr, the modern data was only available for the 2010 summer which would have been insufficient to analyse the frequency distributions. As a result, Birr station was excluded from the modern-day frequency distribution analysis.

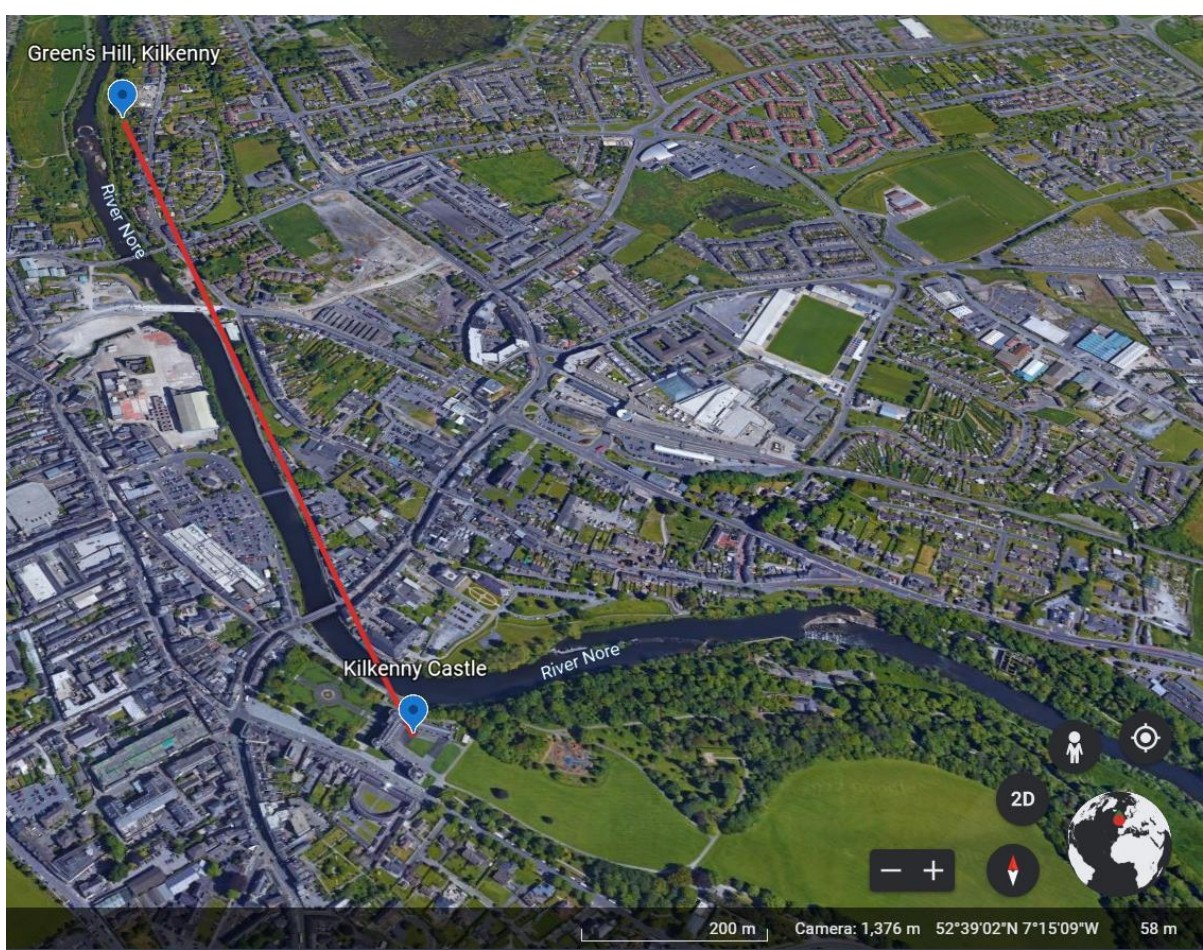

**Figure 6: Illustration of the relative positioning of Kilkenny Castle and Greenshill, Kilkenny (©Google Earth v. 9.154.0.1 (2022)).**

In observing the modern-day differences between Kilkenny Greenshill and 5 of the closest stations with available data in 1887 (Figure 7) several inferences can be made. For the modern-day differences between Markree and Kilkenny Greenshill, 90%

of the distribution lies between a 0°C – 5°C temperature difference (Kilkenny warmer than Markree on average, Figure 6a).

The difference recorded between Markree and Kilkenny Castle on 26th June 1887 was 9.9°C which is a substantial outlier. The bulk of the differences (95%) between Kilkenny Greenshill and Sheffield (2010-2017) lie between -5°C and 5°C (broader because of the far greater distance between the sites). Despite this increased dispersion of modern era differences, the difference between Sheffield and Kilkenny Castle on 26th June 1887 of 9.4°C again indicates a substantial outlier relative to modern inter-site characteristics. Only a handful of higher differences have been reached within the recent era. For Kilkenny Greenshill

and Phoenix Park (2013-2019), 95% of the data lies between -2.5°C and 2.5°C with Kilkenny Greenshill generally measuring slightly higher temperatures than Phoenix Park. The difference on 26th June 1887 between Kilkenny Castle and Phoenix Park at 9°C, lies entirely outside the distribution of modern inter-site behaviour. The highest recorded temperature difference in the modern era was 7.5°C, indicating the currently recognised heat record to be an extreme anomaly. Most (95%) of differences between Kilkenny Greenshill and Roches Point (2010 -2019) lie within a range between 0°C - 5°C, showing that here again

Kilkenny Greenshill generally measures higher temperatures than Roches Point. The temperature record of the 26th of June 1887 was 6.6°C warmer at Kilkenny Castle than Roches Point and lies within the upper 2% of the differences. Similarly, the difference between Armagh and Kilkenny Greenshill (2010 - 2018) shows that 90% of the data lies within -2.5°C and 3°C. Here, the temperature difference of 6.4°C between Kilkenny Castle and Armagh also lies within the upper 2%, showing that such a temperature difference is unlikely to occur but not impossible.

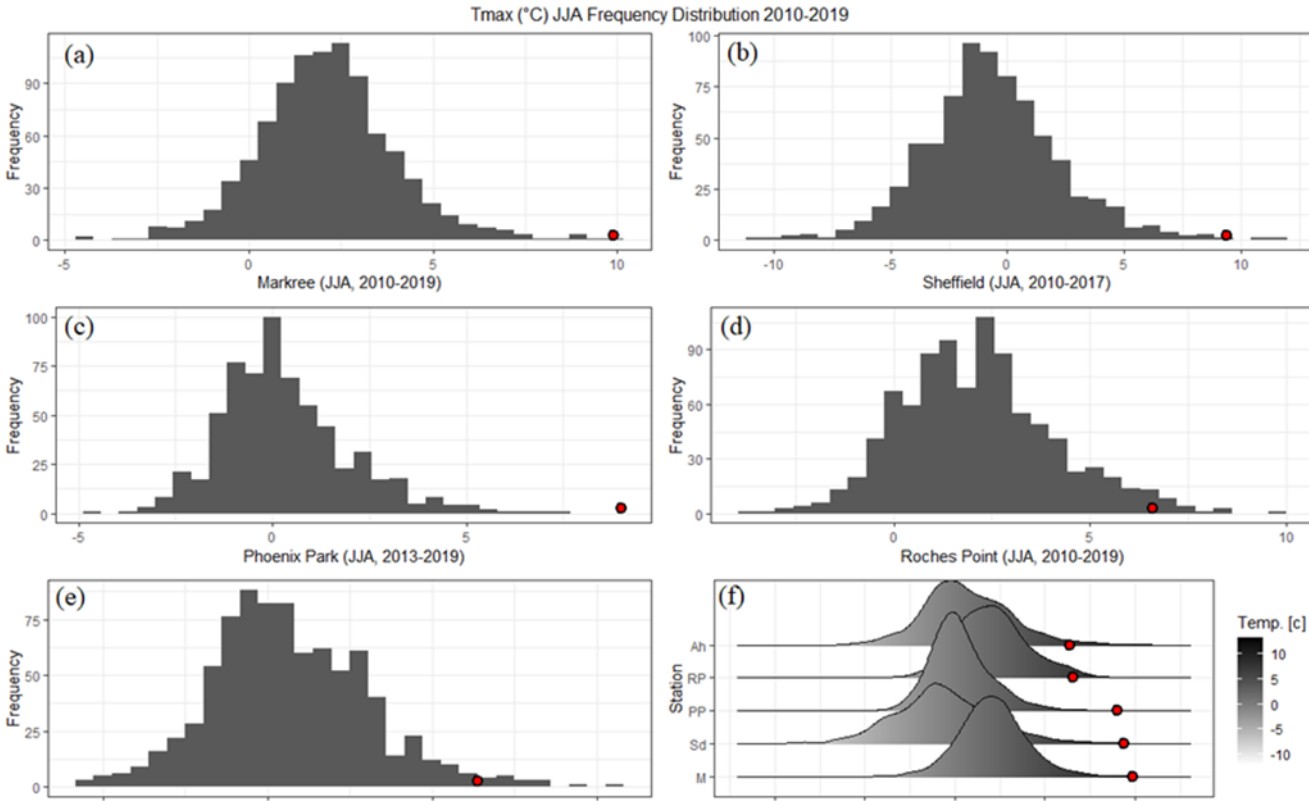

**Figure 7(a-e). Modern day distribution for the JJA season differences between the maximum temperature of Kilkenny Greenshill and the five stations where the data was most available and (f) and amalgamation of all of the stations for comparison purposes. Shown in red is the difference reported on June 26th 1887 between each station and Kilkenny Castle.**

The June 26[th] 1887 difference between Kilkenny Castle and all surrounding sites is highly anomalous being in the extreme upper tail at Markree, Sheffield, Roches Point and Armagh and entirely outside the distribution for Phoenix Park. Differences overall are highly anomalous in a broad arc from SSW through N to due E. While there is a change in circulation via advection of a northerly airmass this occurred on June 27th (Figures 3 and 4), there is no obvious active frontal system that may lead to

a multi-degree thermal gradient between Kilkenny and its neighbours that would support such an extreme departure from all neighbouring sites. The slack pressure gradient also does not support such a strong temperature gradient between Kilkenny and neighbouring sites. While many of the Irish stations are coastal, Birr (for which unfortunately a modern data comparison was not possible) is not and would have been less impacted by any sea breezes that may have set up than Kilkenny. Overall, therefore, while June 1887 undoubtedly was an unusually hot month and there is strong support for a build-up of heat breaking on or around June 26$^{th}$/27$^{th}$, the very anomalous differences compared to modern behaviour argue strongly against the validity of the reported Kilkenny Castle record value. The question therefore is what value may constitute the 'true' Irish national heat record and when did it occur?

## 4. Consideration of Alternative Candidate Record Events

Table 2 noted a number of other candidate dates and locations in which the reported temperature exceeded 32°C. Unlike the Kilkenny Castle value, many of these observations have been made as part of long-term series that are digitally available. These permit, in addition to the analyses performed for Kilkenny Castle, a consideration of the evolution of the station series around the event, including for some cases the hour-by-hour evolution of temperatures on the day. Also, at least for more recent events there is a denser neighbour series network available to perform the neighbour-based comparison and the difference series can be calculated directly rather than via a replacement site as is the case in Kilkenny. The relative recency of the events also means there is better metadata, and the records are generally made in a more standardised and uniform manner, consistent with WMO guidance on methods of observation (WMO, 2018). However, first it is necessary to consider the viability of the other records unearthed by Mateus et al. (2020) many of which pre-date the Kilkenny Castle record.

### 4.1. Early period candidate record high temperatures

Mateus et al. (2020) have digitised a range of early Irish meteorological stations and made these data available. In constructing Table 2 the majority of the remaining observations above 32˚C arise from this source, including an observation of 33.5˚C from

Phoenix Park, Dublin in July of 1876, which is even warmer than the Kilkenny Castle observation, and a set of observations made in the 1850s

### 4.1.1. 1850s candidates

The suite of observations made in the 1850s, with the exception of the RCS observation in 1856, preceded the formation of the UK Met Office in 1854 (at the time the Republic of Ireland had not gained independence from the UK). Metadata on who exactly took these observations is incomplete but suggests they were undertaken by military engineers to whom it is assumed some degree of training had been given. This period precedes efforts to shield instruments using a Stevenson Screen (Stevenson, 1864) and instruments may typically have been housed on north facing walls or placed in ventilated rooms (Parker, 1994). There is substantial literature pointing to potential biases in these early records with a tendency for summertime maxima to be overestimated (Parker, 1994, Camuffo, 2002, Böhm et al., 2010, Trewin, 2010, Brunet et al., 2011). Specifically, modern-day comparisons either of original instruments and exposures (Böhm et al., 2010) or reconstructed instruments and exposures (Brunet et al., 2011) with standard Stevenson Screens highlight important potential biases in summertime daily maximum temperatures. These are particularly marked for mid-latitude north wall exposures that would, especially if not oriented to true north, catch significant solar radiation in mid-summer (Parker, 1994, Böhm et al., 2010).

A further concern over the validity of several of these 1850s high temperature records relates to their geographical situation. A reading of 32.2°C on Scattery Island, Clare, an Island off the west coast of less than 1km by 1km seems implausible when climatological sea surface temperatures are in the high teens. Markree in Sligo also appears multiple times. While it is possible in theory to have a high temperature at these stations given the presence of a long upstream land passage of air combined with an offshore wind, as a near-coastal location in the NW of Ireland it is difficult to envisage how temperatures as warm as 32°C or higher could be attained on such a regular basis. They also are very much bunched around late June which may indicate the presence of radiative effects around the solstice upon this measurement series or something as simple as annual leave measurements being taken incorrectly by a substitute observer. Kilrush, in Clare is similarly coastal and similar questions would pertain around how plausible such extreme heat was so close to the Atlantic Ocean. The same concerns pertain to

Dunmore East which is on the Irish Sea coast of SE Ireland. The Royal College of Surgeons in Dublin is relatively close to the quay so, again, it is questionable whether such a warm temperature could be attained even allowing for urban heat island effects.

The 1850s candidates also are sometimes associated with implausible synoptic situations according to NOAA 20CRv3 MSLP and 850-hPa Temperature reanalysis reconstructions (Figure 8). Even in the 1850s the available pressure constraint is sufficiently robust over NW Europe to provide a robust synoptic scale reconstruction. The three synoptic charts in 1851 (Figure 8) are associated with high pressure and a slack flow of air from the continent which would, generally, be associated with climatologically hot conditions. Of note is that over 28th/29th of June 1851, Kilrush, Markree and Dunmore East all reported temperatures in excess of 32°C. Combined with the synoptic chart reconstructed from 20CRv3 there is little doubt that 28th/29th June 1851 was, indeed, a heatwave event across the island of Ireland. Newspapers at the time reflected these conditions reporting on the 'hot' and 'oppressive' weather that was occurring in Ireland (Freemans Journal, 1851; Leinster Express, 1851).

The remaining dates in the 1850s are associated synoptically with conditions that would tend to be climatically normal or even below normal temperatures. The 27/06/1852 chart has a low pressure to the north of Ireland with a slack westerly flow which would be an onshore wind at Markree, Sligo, and taken together with the depressed 850hPa temperatures would be completely inconsistent with a temperature in excess of 32°C. The 26/06/1853 event, again for a temperature at Markree, Sligo, is represented by a south-westerly flow, which is grossly inconsistent with temperatures as high as the low 30s in this location. The situation on 29/06/1854 is of a low pressure centred over England with a northerly flow, again inconsistent with temperatures in excess of 32°C at Markree. 850hPa temperatures are also notably depressed on this date casting considerable further doubt on the verity of this record. The slack pressure gradient on 03/08/1856 leads to potentially a slight easterly onshore wind inconsistent with a high temperature at the Royal College of Surgeons in Dublin.

Taken together, the uncertainty relating to early instrumental bias, climatological locations, and in many cases lack of substantive synoptic situation support mean that the 1850s values contained in Table 2 almost certainly should be precluded as robust candidates for record temperatures. That is not to say that, at least in some cases, the values may not be correct or at the very least indicative of, climatologically speaking, extreme heat. But Markree in particular, which occurs the most

frequently, has values that are often not supported at all by the synoptic situation.

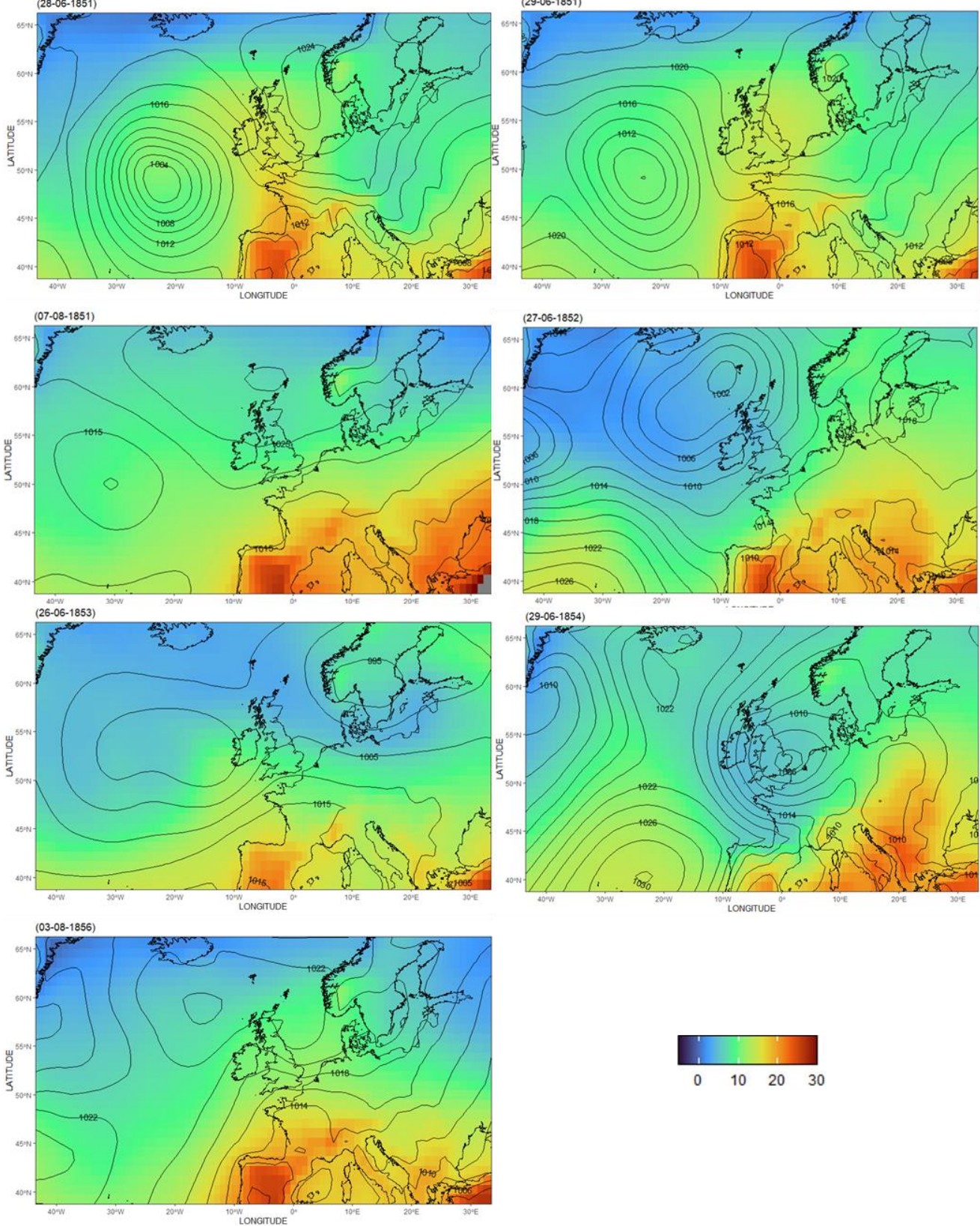

**Figure 8. NOAA 20CRv3 MSLP and 850-hPa Temperatures on the dates of the possible heat records in the 1850s arising from Mateus et al. (2020) recent data rescue and as documented in Table 2. 20th Century Reanalysis V3 data provided by the NOAA/OAR/ESRL PSL, Boulder, Colorado, USA, from their Web site at https://psl.noaa.gov/data/gridded/data.20thC_ReanV3.html. The domain of latitude 35N-70N and longitude 45W-30E was chosen to provide a synoptic view, with Ireland centred to provide an overview of the surrounding regions. All maps correspond to 1500 UTC which is the approximate expected time of daily maxima in Ireland (corresponding to c. 14.30 LST).**

### 4.1.2. The Phoenix Park value of 33.5°C

The Phoenix Park site is the longest running contiguous site in the Republic of Ireland. Originally run by the Ordnance Survey, then the military, it is now maintained by Met Éireann. It is a little unclear exactly how the potential record on 16th July 1876 was measured. Mateus et al. (2020) mentions "Drawings when no station photographs are available for the early 19th century, for example Cameron (1856) of the thermometer screen at Phoenix Park Dublin, are furnished". In 1879 there is the first mention of measurements by a Stevenson Screen "quality of the record improved and since 1879/1880, when the thermometers were housed in a Stevenson screen, the data may be considered to have a high level of accuracy and reliability [sic]" (Irish Meteorological Service, 1983: 4). It is hence probable, but not certain, that the measurement was made by a thermometer housed in something other than a Stevenson Screen, although exactly what is hard to ascertain precisely. The site is about 5km inland from Dublin Bay such that, in the absence of a sea-breeze, climatologically speaking, a temperature in the 30s could potentially be attained. The synoptic situation as reconstructed by NOAA 20CRv3 is supportive of very warm conditions with a near-stationary high pressure of 1025hPa centred over the British and Irish isles (Figure 9). Preceding days had had a gentle flow of air from the near continent / Iberian peninsula. The 850hPa temperature evolution (Figure 10) is also highly supportive of very warm temperatures – more so than for the Kilkenny Castle record (c.f. Figure 5). The 850hPa temperatures are consistently in the high teens to low twenties and peak on 16th, consistent with the date of the recorded maximum. The peak results from the advection of very warm air over several days from the continent. There is thus clear synoptic support for very high temperatures at this time.

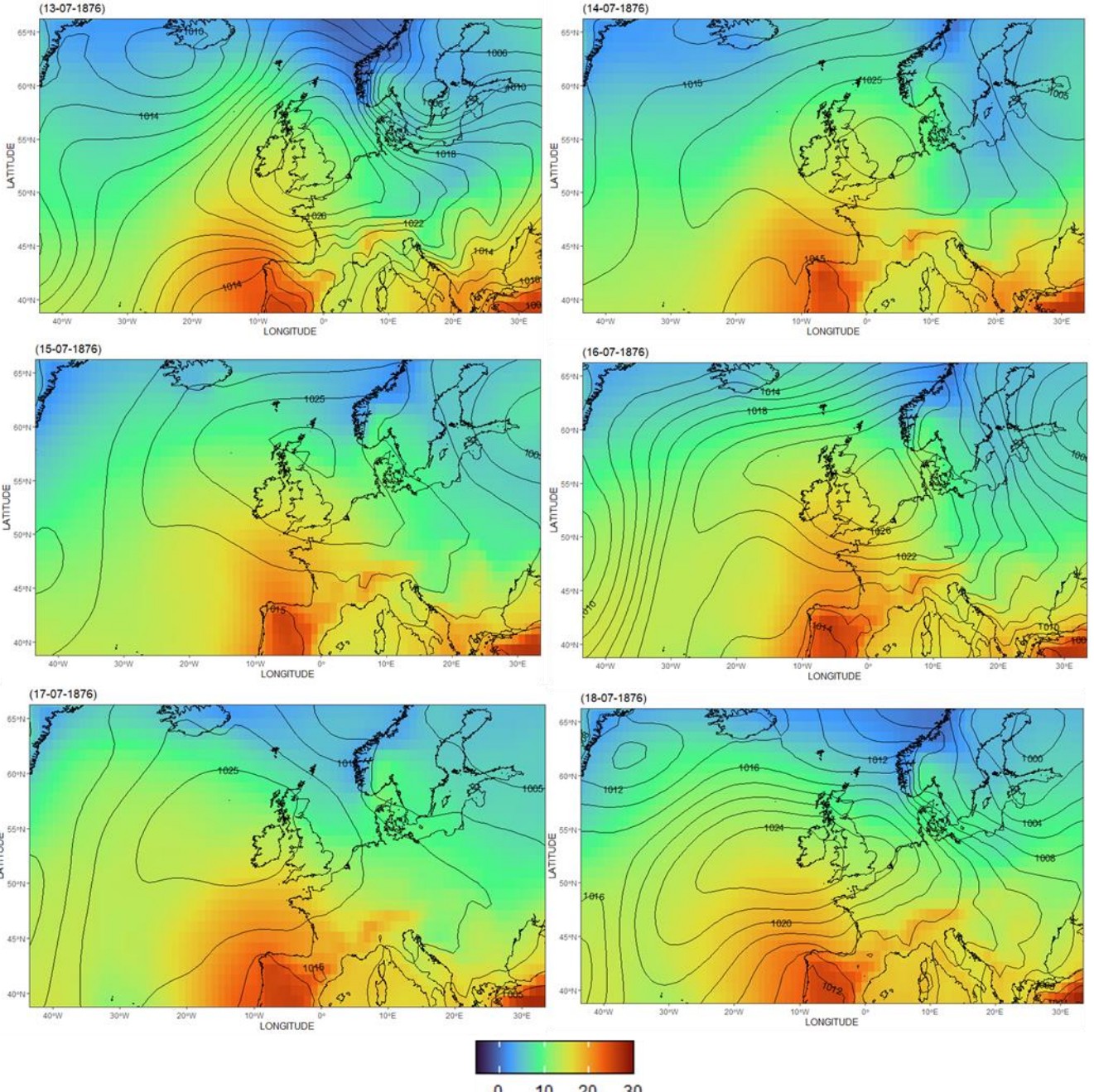

**Figure 9. NOAA 20CRv3 MSLP and 850-hPa Temperatures evolution around the Phoenix Park observed value of 33.5°C in 1876 on 16th July. 20th Century Reanalysis V3 data provided by the NOAA/OAR/ESRL PSL, Boulder, Colorado, USA, from their Web site at https://psl.noaa.gov/data/gridded/data.20thC_ReanV3.html . The domain of latitude 35N-70N and longitude 45W-30E was chosen to provide a synoptic view, with Ireland centred to provide an overview of the surrounding regions. All maps correspond to 1500 UTC which is the approximate expected time of daily maxima in Ireland (corresponding to c. 14.30 LST).**

375

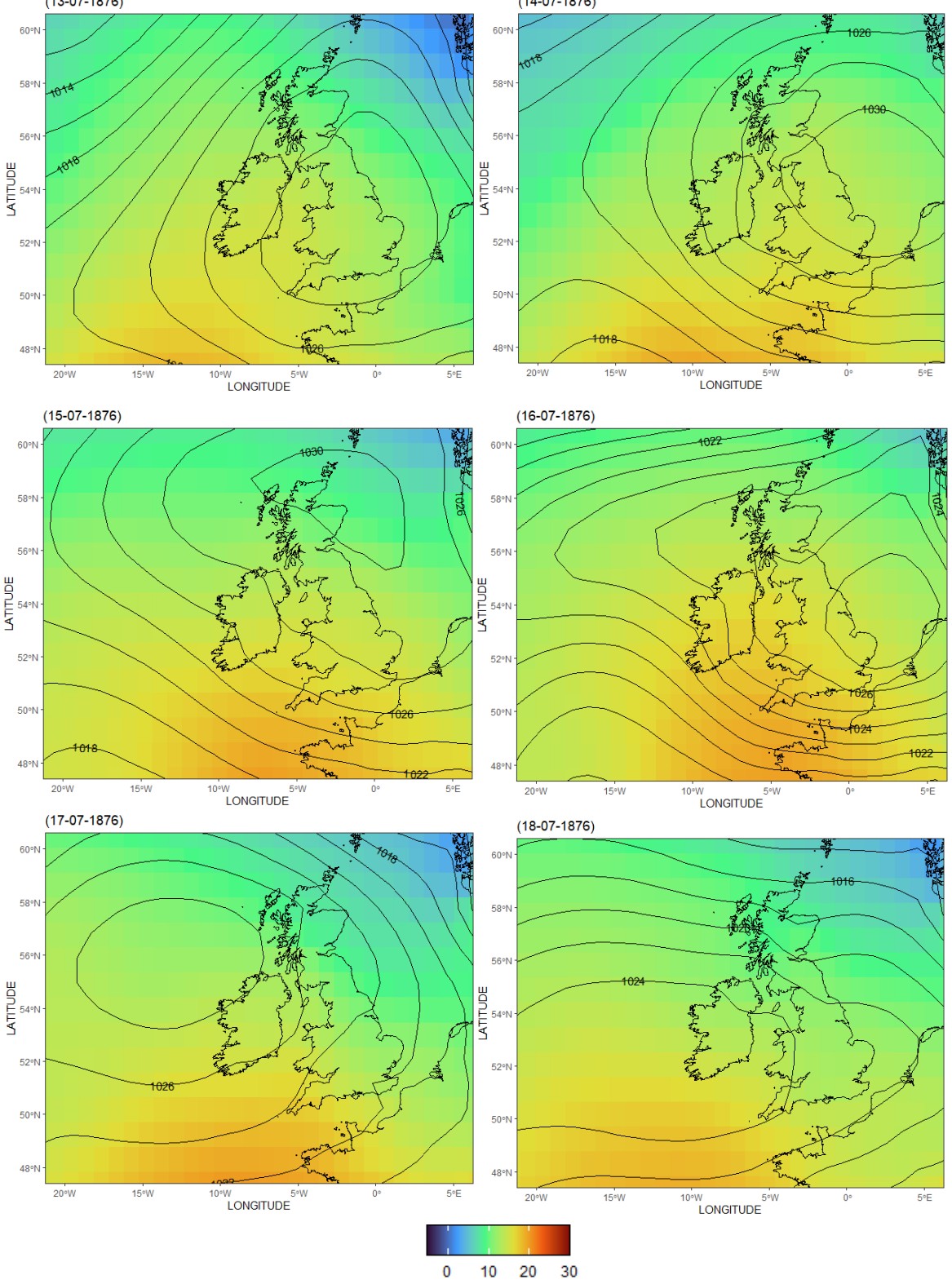

**Figure 10. As figure 9 but for zoomed in data to the British and Irish isles.**

Having discounted the long-standing Kilkenny Castle national record in Section 3 and, in Section 4.1, ruled out as sufficiently robust any values from the 1850s, this leaves the Phoenix Park temperature a full 1°C warmer than any other remaining candidate value in Table 2. Most of these remaining candidate values are from far further inland where such high temperatures *a priori* would be more easily attainable. When analysed against other Dublin city temperature observations for the month of June 1876 (Figure 11) it can be seen that, while a similar temperature evolution is mostly followed throughout the month, the Phoenix Park station is consistently hotter than the other Dublin stations (except for the 3rd and 26th day) by between 1 and 2 degrees. Dixon (1953) reported a sudden and rapid temperature high just a few years earlier, on the 21st July 1868; "*the temperature in Phoenix Park one day during July shot up to 88.4 degrees F.* (31.3°C), *a record which has yet to beaten*". At this time, the Botanic Gardens similarly recorded a very hot July, however the temperature recorded on the 21st of July is 81 degrees F (27.2°C), several degrees below the Phoenix Park recording despite their close geographical proximity.

It is clear from the 20CRv3 reconstruction and the surface temperature recordings shown in Figures 9 and 10 that the synoptic situation in July 1876 supports very warm conditions. However, the near constant systematic elevation of the temperature readings in Phoenix Park relative to other stations casts doubt on whether it was reporting the actual surface temperature at the time. Phoenix Park is in an elevated position so on lapse rate basis alone would be expected to record slightly lower and not higher temperatures than nearby sites. While the value at the Phoenix Park cannot be discounted there is equally no obvious way to sufficiently robustly confirm the observation for it to be adequately verified as a national heat record.

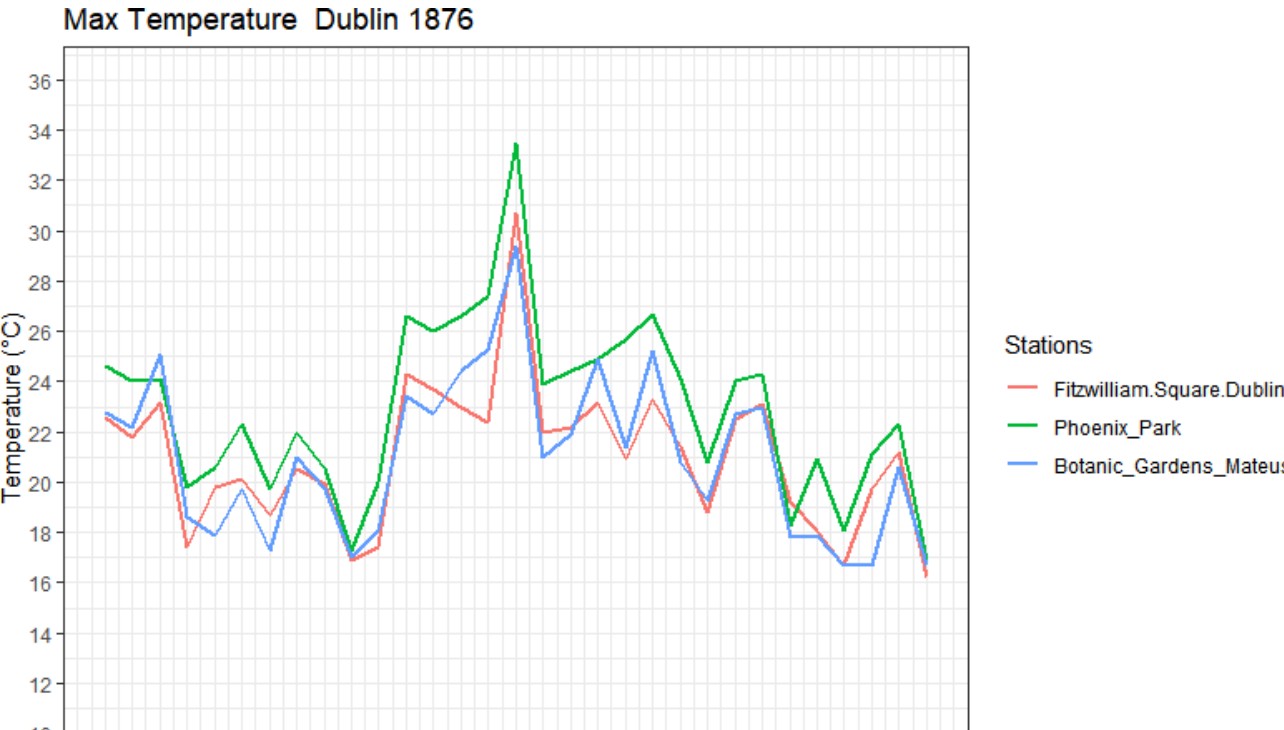

**Figure 11: The maximum surface temperatures of all available stations in Dublin City during June 1876.**

### 4.2. Candidates since the beginning of the 20th Century

The remaining candidate events in which the reported temperature attained or exceeded 32˚C are all associated with synoptically broadly similar conditions. High pressure in each case is situated over or to the north / north east of the island of Ireland, often with weak advection of air from the near continent (Figure 12). For all cases the 850hPa temperatures are elevated

relative to seasonal expectations with 850hPa temperatures in the high teens to low twenties. The synoptic situations thus do
       not call into immediate question any of the remaining candidate values realism.

# NOAA 20CRv3 MSLP and 850hPa temperatures

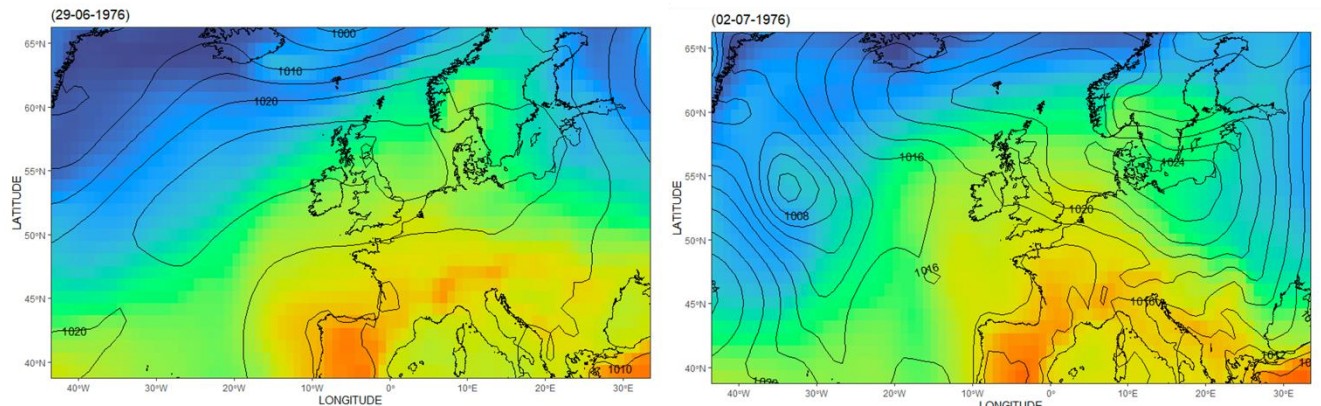

## ERA5 MSLP and 850hPa temperatures

Boora is the hottest value of these remaining candidates at 32.5°C, attained on 29/06/1976. Boora is situated in the centre of

the island of Ireland, which means it is highly unlikely to suffer from marine influences which may cap diurnal maxima closer to the coast. The summer of 1976 was famous across the British and Irish Isles for its drought and hot summer following on from 18 dry months previously (Noone et al. 2017; O'Laoghog, 1979). Therefore, the partitioning of heat between sensible (temperature) and latent (evaporation) terms would have been climatologically skewed toward sensible heating. The value at Boora furthermore is situated within a string of very warm days with many of the daily maximum

observations exceeding 30°C.

The hot and dry conditions were widely reported in newspapers at the time; The Irish Press reported on 24ᵗʰ June 1976 '*The Irish have been greeted by a heatwave which saw the temperature hover around the 88 degree (31.1°C) mark for a long time*' (Redmond, 1976). On 28ᵗʰ June 1976, the Irish Press reported on the weather in Britain and Europe: '*you cannot expect to*

*share the 90 degree (32.2°C) heatwave that has had Britain sweltering for days. But there is a chance you will soon be basking in a purely Irish hot spell, with temperatures reaching 85 degrees F (29.4°C). Britain, because of its proximity to the Continent and its distance from the Atlantic, is enjoying the effects of an anti-cyclone which is almost stationary over Europe...According to the Dublin Met Office, this could mean an Irish heatwave within the next few days, with temperatures rising from the present 60 - 70 degrees F. (15.5 – 21.1°C) to 86 degrees F (30°C)*' (Cahill, 1976).


A comparison to neighbouring sites (Figure 13) for this candidate event benefits from the standardisation of meteorological observations during the 20ᵗʰ Century and the much denser network of Met Éireann sites since the mid-20ᵗʰ Century. Comparisons can now be made to sites at tens of kilometres distance rather than some hundreds of kilometres. These elevated

temperatures, experienced at Boora, were also experienced at the nearby stations which sit within tens of kms of the Boora site
(Table 3).

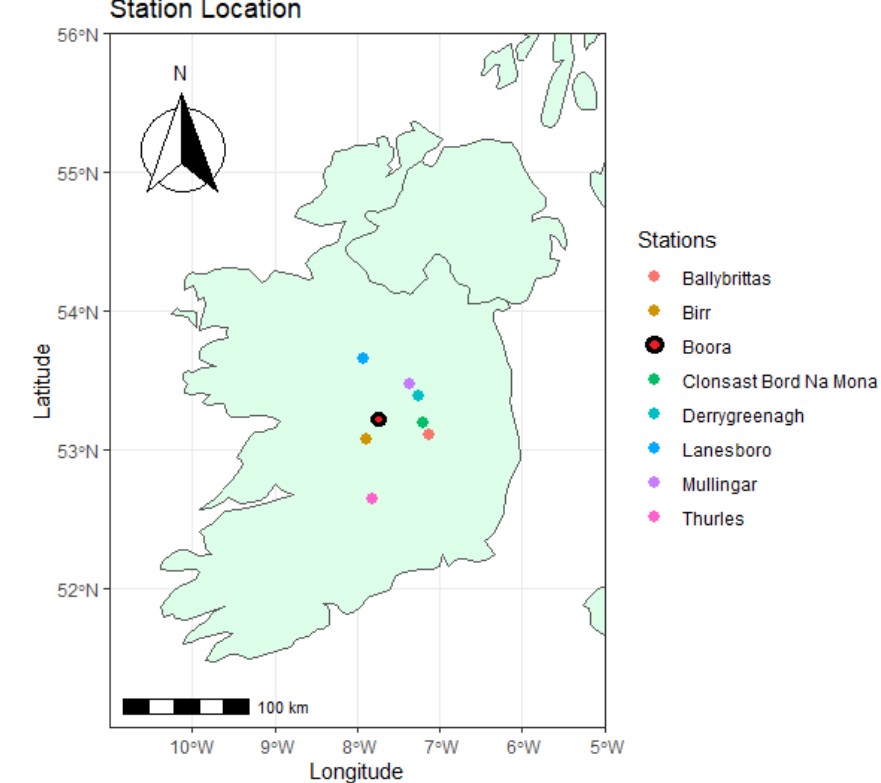

**Figure 13: Map of stations closest to Boora considered in this analysis**

| Station name | Distance from Boora (km) | Latitude | Longitude | Elevation (m) | Observed Maximum Temperature (°C) | Difference to Boora (°C) |
|---|---|---|---|---|---|---|
| **Boora** |  | **53.22** | **-7.72** | **58** | **32.5** |  |
| Birr | 19 | 53.09 | -7.89 | 72 | 31.2 | -1.3 |
| Mullingar | 41 | 53.31 | -7.21 | 101 | 28.8 | -3.7 |
| Clonsast Bord Na Mona | 35 | 53.11 | -7.12 | 73 | 30.8 | -1.7 |
| Derrygreenagh | 36 | 53.21 | -7.15 | 90 | 28.7 | -3.8 |
| Ballybrittas | 41 | 53.06 | -7.08 | 90 | 32 | -0.5 |
| Lanesboro (Doire Dharog) | 53 | 53.67 | -7.93 | 49 | 30.2 | -2.3 |
| Thurles (Sugar Factory) | 58 | 53.65 | -7.82 | 101 | 31.6 | -0.9 |

**Table 3. Comparison between Boora, Offaly maximum temperature on 29/06/1976 and neighbouring stations within 60 km radius. All data arise from Met Eireann.**

Repeating the methodology of the Kilkenny neighbour-based analysis but using Boora and nearby stations for a period of up to 20 years either side of the record, the observation in question sits well within the distributions of expected differences in

daily maxima over summer (Figure 14). In the modern-day differences between Boora and its seven neighbouring stations, most of the inter-site distribution lies between ± 2°C temperature difference and almost all daily values lie within ± 5°C. For Boora and Derrygreenagh, most of the data lies between -1.5°C – 2°C with Boora typically measuring slightly higher temperatures than Derrygreenagh. The difference on 29[th] June 1976 between the two sites at 3.8°C, lies within the upper 2% of the expected differences but has still been exceeded on several occasions. The difference recorded between Boora and

Mullingar was 3.7°C which falls within the upper 2% of its distribution. The difference between Boora and Lanesboro, at 2.3°C sits well within the distribution of daily differences experienced between the sites. Differences between Boora and Clonsast are generally between -2°C and 2.5°C and the heat record difference on 29th June 1976 was 1.7°C, falling within the upper 10%. Birr was 1.3°C cooler than Boora, sitting well within the range of experienced daily differences. Similarly, the difference to Thurles at 0.9°C almost matches the peak frequency of differences expected between the stations. Lastly, Boora

and Ballybrittas have very strong similarities in the recorded temperature with most of the differences between -2°C and 2°C. The June 29th 1976 difference between these stations was 0.5°C completely within the distribution of modern inter-site behaviour.

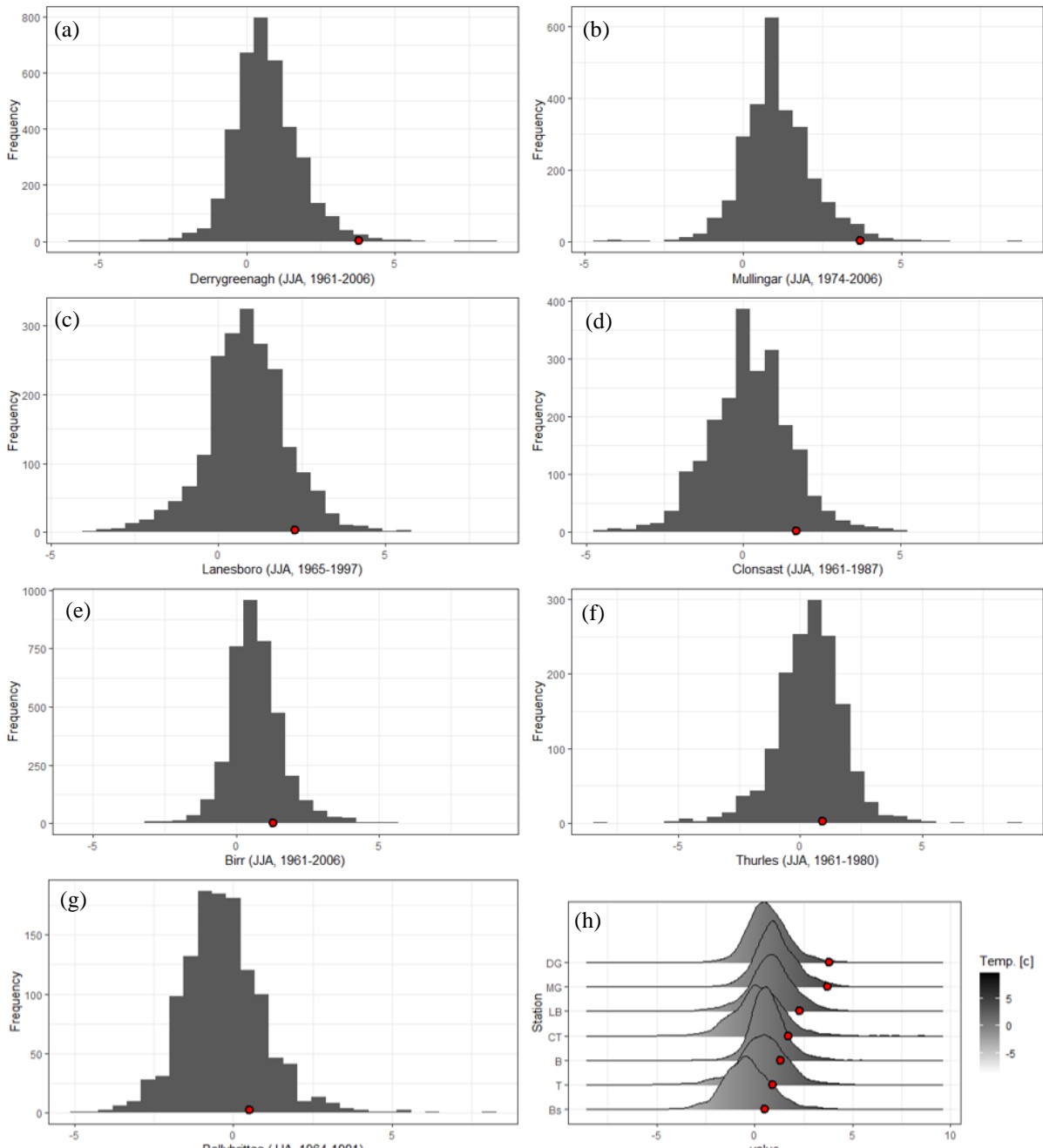


**Figure 14. Modern day distribution for the JJA season differences between the maximum temperature of Boora, Offlay and the seven stations where adequate data was available. Shown in red is the difference reported on June 29th 1976 between each station and Boora.**

A combination of geographical location, the synoptic situation, antecedent conditions, neighbour comparisons and use of
standard modern-day meteorological observations leads us to conclude the Boora observation is likely to be a credible reading. We would thus conclude logically that this represents the highest reliably recorded temperature in the Republic of Ireland since observations began up until the time of the present analysis in 2021. If the observation were to be called into question any of several candidates between 32°C and 32.3°C since 1900 based upon the synoptic situation on each date would be viable alternatives, but we do not consider these in greater depth herein.

**5. Conclusions**

Ireland is now highly anomalous in having its recognised national heat record set in the 19th Century. The record is from a long-closed site in Kilkenny Castle and the associated series is not available, presumed lost, as is the bulk of relevant metadata. The Kilkenny record of 33.3°C in June 1887 exceeds that reached since the beginning of the 20th Century by 0.8°C. Results from our inter-station reassessment and reanalysis comparisons leads us to question if this truly represents the highest *reliably*
measured national temperature for Ireland. Based upon the few contemporaneous station records available on the island of Ireland and at Sheffield and their modern differences to a site close to the original Kilkenny Castle site, our findings cast very considerable doubt upon the record. Differences on the 26th June 1887 are implausibly large and not consistent with the synoptic situation as reanalysed by NOAA 20CRv3. A range of similar concerns including basic physical considerations preclude a number of other 19th Century values in excess of 32°C, including an observation of 33.5°C at Phoenix Park in 1876.


A search of candidate records since 1887 yields that the highest likely defensible observation is 32.5°C recorded at Boora on 29th June 1976. The synoptic situation is consistent with a significant heatwave, the series evolution is realistic and comparison with nearby (within 10s of kms) neighbouring stations does not suggest any substantial issues in the reported reading. We

would thus recommend that the Boora observation be recognised as the national heat record for the Republic of Ireland. Ultimately, however, the official recognition of the national temperature record rests with Met Éireann as the national meteorological service.

It should be noted that following the completion of this present analysis, on the 18[th] of July 2022 Met Éireann recorded a temperature of 33.0°C at the Phoenix Park Station in Dublin. This record is still waiting to be verified by Met Éireann but if correct, would constitute the new national all-time hear record in place of Boora, Co. Offaly.

This analysis only considered the national all-time heat record. Met Éireann maintains all time annual and monthly records, several of which date back to similarly early, pre-standardisation, measurements (Met Éireann, 2020). Most notably the all-time cold record is -19.1°C, recorded in Markree, Sligo in January 1881 which is close to the Atlantic coast in NW Ireland. Reassessment of these records using a range of techniques such as those used here would likely be valuable.

It is probable that Ireland is far from alone in having national meteorological heat, cold, rainfall and other records that may benefit from a reassessment using these new techniques pioneered by WMO record assessment teams. These national all-time meteorological records are important in media and the general public discourse around both weather and climate change. They are also important in the context of monitoring long-term changes in climate extremes and their attribution. It is therefore important that the verity of these records be assured. The techniques and approaches used herein are broadly transferable to any similar reconsideration of meteorological records in other jurisdictions.

**Code availability.** The data and code that support the findings of this study are available through: https://github.com/katherinedooley/Reassessing-Ireland-s-Hottest-Temperature-Record

**Competing interests.** The authors declare that they have no conflict of interest.

**Author contributions.** SOK, CC, EC and KM conducted a primary investigation and provided background information on the study. KD, CK, NS, and JKD conducted an inter-station series assessment to assess the plausibility of the reported record.

DC, NM, JD, TM created and analysed reanalysis weather maps for the top recorded hottest years 1887, 1995, 2006 and 2018. RS, EG, JC explored the plausible national heat records if Kilkenny 1887 record was to fail. SN and CM provided data and support along with contributions from Met Éireann. KD, CK and PT prepared the paper with contributions from all the co-authors.

**Acknowledgements.** The authors would like to extend our gratitude to ICARUS and the Department of Geography in Maynooth University for their help on this assignment for the MSc in climate change. The authors would also like to thank the reviewers of this paper, Chris Folland and one anonymous reviewer for their time and expertise given in improving the paper.

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
