# Peer review of "Reassessing long-standing meteorological records: an example using the national hottest day in Ireland"

_Climate of the Past, 2021_

## Author Comment (AC1)

**Reviewer 1**

General

This paper makes a well-argued case that the record Irish (Eire) temperature should be the record for Boora in 1976.and not Kilkenny in 1887. However, I have a significant suggestion that could make this finding either stronger or weaker, or just possibly, change the decision. This involves considering the evidence from the several Figures in the paper like Figure 4. Here 500mb heights are shown in addition to the very useful pressure at mean sea level (PMSL) maps. It would be considerably more useful to replace 500mb heights, difficult to interpret in the context of this paper and not made much of anyway, by a relatively low-level reanalysis temperature field that is unlikely to be corrupted by a bad surface air temperature observation. I come back to this in the specific comments below

The other main comment is that aspects of the presentation should be improved, particularly many of the remaining diagrams. Nevertheless, the text is generally well written. I recommend this paper be accepted subject to significant/major revision.

*We thank the reviewer for the time taken to review our manuscript and their very useful feedback and helpful suggestions. We respond to specific points as they arise below and will of course attempt to improve the aspects of the presentation mentioned above (and below).*

Specific Comments

1. Lines 30- 62 of Introduction and/or Table 2. Somewhere about here it would be very useful to add the latitude, longitude and height location of Kilkenny Castle. In Table 2 it would add useful information to add these details for all the other stations shown – much as done in Table 3.

   *Both will be added in our revisions. With regard to table 2- Lat, Long and height ASL will be added for those cases where the exact location is known.*

2. Figure 1. It is not so easy to see on a printed copy where Kilkenny Castle is located on either map. Perhaps an arrow and superimposed text of suitable size saying "Kilkenny Castle" could be added.

   *Thanks. We will edit this figure to improve this aspect.*

3. Figure 2. The coloured dots would be easier to see if the green background was much lighter.

   *Figure will be amended as requested. We agree this will improve presentation.*

4. Figure 3. This diagram is not very clear on a printed copy, mainly because of the grey background. This can be fixed by making the graph lines thicker. The key is odd: the heading should be "stations" not "variable" and Kilkenny has a coloured line in the key, even though there are no data for this date, except the single point highlighted in the text. On the left, the label should add "degrees" before "Celsius". In the caption "air" should be added after "surface".

*The figure will be revised as indicated and we thank the reviewer for these helpful recommendations.*

5. Figure 4 and similar Figures. This is my most important comment. On the main Climate of the Twentieth Century Reanalysis v3 NOAA web site atmospheric temperature analyses are available in ensemble mean form for 8 times of the day for each day until 2015. I strongly recommend that 500hPa height be replaced by an appropriate relatively low-level temperature (850hPa temperature or lower, as long as not significantly affected by station biases). This would be done (probably) for 1500 hours for each sub-diagram shown in Figure 4 and the similar figures. Ensemble mean PMSL would need to be be plotted for the same time of day for consistency. This would add genuine new information about the relative likelihood of the hottest temperature extremes. This type of information could become an important part of the evidence for daily surface air temperature extremes more generally. As far as I can see, appropriate sub-daily maps can be easily plotted for such data using the on line facility at https://www.psl.noaa.gov/data/20thC_Rean/ . After 2015, similar data should be available from ERA5.

*We thank the reviewer for this recommendation, and we will integrate this into our revised manuscript.*

Figure 5. On a printed copy, the red circle is less clear on some sub-diagrams than it should be. These are good diagrams otherwise. It might be useful to add combined simultaneous differences for all stations in a sixth panel -currently blank.
*Thanks for checking and highlighting. We will make this addition to Fig 5.*

6. Figure 7. Where relevant, I have the same comments about the graphs as for Figure 3.
*We will make this adjustment in the revised manuscript.*

7. Figure 10. Same comments about the red dots as for Figure 5.

8. *We will make this adjustment in the revised manuscript.*

Minor comments

1. At Line 241, Parker (1994) is missing from the Reference list.

*We will add, thank you for catching that.*

---

## Author Comment (AC2)

This analysis reassess the 1887 high temperature record of Kilkenny, Ireland, using different datasets including the early-year records, current observations and reanalysis product, and concludes that this record and other early observations of extreme high temperature are not sufficiently reliable. The authors instead suggest the 1976 heat record at Boora as the highest national temperature measurement. The analysis is interesting, and the conclusion is also more or less convincing. This is not an easy work anyway. One of my concerns is the locality of the study, and thereby the narrowness nature of its significance if it is published in the CP. In addition, I find a few of other issues which demand further clarification.

*We thank reviewer 2 for the time taken to review our manuscript and the positive response provided. We believe that our manuscript is of interest to the readers of CP given the methods and multiple approaches adopted to evaluate a questionable historical extreme, therefore going beyond local interest. We will of course endeavour to provide further clarification as necessary.*

1. The Kilkenny observational site is located in the centre of Kilkenny Town. Was it also located in the town centre in 1887? What was the extent and population of the town in the early time? If the town has not changed much in urban area and population, would the central location of the observation in 1887 be more productive to the higher temperature records compared to other observational stations in the nearby cities and towns of similar size?

   *The Kilkenny observational site is currently located in a private residence, which is believed to be a different location to the recording of 1887. However, the specific location of the 1887 record is not known within the Kilkenny Caste grounds.*

   *In 1841, the population of Kilkenny City was approximately 19,071, decreasing to 15,257 in 1851 (McDill, 2021). In 1891, the population total of the county of Kilkenny was 87,261 (CSO, 2004). According to the 2016 census, Kilkenny City has a population of 25,512 (City Population, 2021). Therefore, it can be stated that the population has grown in Kilkenny City by over 10,000 since the mid-1800s. Since the 1800s, new buildings and developments have occurred in Kilkenny City. Although many historic buildings remain, ultimately the city is not the same as it was in 1887.*

   *The closest weather station to Kilkenny Castle is Kilkenny Greenhills, which became a climatological station with daily observations from manual instruments being submitted to Met Eireann in 2010 (Kilkenny Weather, 2019). The maximum temperature recorded at this station since 2010, was 29.6 °C in 2013 and 2021. Other weather stations in Kilkenny do not record daily maximum temperatures. A figure has been added to illustrate the relative locations of the castle and the newer site.*

   *We will add this additional insight to the revised paper.*

2. The application of 20CR to show the synoptic situation seems not so persuasive, I feel. It indicates a favorable weather condition for the extreme hot and dry event, but when the authors said that the condition is not uncommon in the summer season over

Ireland, they did not exhibit an example of the synoptic situation at present. Is there any similar or even more extreme high pressure system over the study region during the same length of periods in modern time?

*Thank you, in line with the response to Reviewer 1 we will update this analysis to replace with MSLP and 850hPa temperature reanalysis.*

3. The exercise to compare the JJA season differences of maximum temperature between Kilkenny Greenshill and the four stations is convincing procedure. However, the representativeness/proximity of Kilkenny Greenshill to Kilkenny Castle would be important. The authors did not well document this. It would be good to at least give a large-scale map showing the specific locations of the two sites. Besides, all the data used should be evaluated for their quality and homogeneousness.

*Thanks for spotting this. We will add a map showing the proximity between these two stations as a new figure.*

4. Given Kilkenny Greenshill could well represent Kilkenny Castle, there are other issues that should be clarified. 1) Birr seems mostly close to Kilkenny Greenshill, with both being inland stations, but its modern data was not shown, why?

*Thank you, we have now provided information in the paper as to why Birr was excluded from the modern-day frequency distribution. The modern data was only available for the 2010 summer which would have been insufficient to analyse the frequency distributions. As a result, Birr station was excluded from the modern-day frequency distribution analysis. This detail will now be made clear in the revised manuscript.*

2) For the two sites (Roches Point and Armagh) which is located inland or more closely to Kilkenny, the differences of Tmax between 1887 Kilkenny Castle record and those of the two stations are within 99% of the modern data, which means that the differences are not impossible. If the reference sites are closer to Kilkenny, would the possibility for the 1887 Tmax differences to occur be higher? –

*The modern-day frequency distribution shown in Figure 5, does provide a possibility for the 1887 heat record of Kilkenny to have occurred however the frequency distribution also suggests that the 1887 heat record would be an extremely unlikely occurrence. Furthermore, the June 1887 Tmax representation of the six selected weather stations shown in Figure 3, clearly has the Kilkenny heat record displayed as an outlier in comparison to its reference sites, If one or two sites being compared were at 99% of distribution that would be one thing, but for all comparator sites in all directions to be either at the limit of or entirely outside the expected distribution is quite another. It is both the extreme nature of individual pairwise comparisons, but more that in all directions these fall either outside or right in the extreme tail that raises serious questions around this possible record event.*

5. Finally, I would suggest that all the figures be improved for their quality and clarity.
*Noted. All figures will be improved in line with suggestions from both reviewers.*

---

## Author Response (AR1)

Dear Dr Dooley and co-authors

Thank you for posting your responses to the interactive comments. I now invite you to submit a revised version following the suggestions of the two reviewers according to the outlines provided by you. While all reviewers comments are relevant, I suggest that you do elaborate on the following comment from ref #2: "One of my concerns is the locality of the study, and thereby the narrowness nature of its significance if it is published in the CP". In other words, your study would benefit from being set in a larger context (e.g. importance of meteorological data rescue).

Good luck with the revision

Hans Linderholm

*Dear Dr. Linderholm*

*Thanks for the opportunity to proceed to this next stage of the process. Also our thanks to the two reviewers for constructive reviews. We believe that this analysis will be of interest to the journal readership and that the comments of the reviewers and yourself will improve this aspect of the work.*

*With regard to the locality and potential narrowness of the study, the text has been reworked to stress the importance of the reanalysis of historical records more generally and with a specific focus on those records that are of long—standing and precede efforts to standardisation of instruments and methods of observation. It is this latter aspect which leads us to believe that CP is an appropriate fit for this analysis. We now better highlight the potential value in reanalysing other national meteorological records and better outline how the techniques and approaches used in the paper are broadly transferable to any similar reconsideration of meteorological records in other jurisdictions. We thank you for your feedback on this.*

*We have included detailed responses to each of the reviewer comments, outlined in blue below.*

**Reviewer 1**

General

This paper makes a well-argued case that the record Irish (Eire) temperature should be the record for Boora in 1976.and not Kilkenny in 1887.  However, I have a significant suggestion that could make this finding either stronger or weaker, or just possibly, change the decision. This involves considering the evidence from the several Figures in the paper like Figure 4. Here 500mb heights are shown in addition to the very useful pressure at mean sea level (PMSL) maps. It would be considerably more useful to replace 500mb heights, difficult to interpret in the context of this paper and not made much of anyway, by a relatively low-level

reanalysis temperature field that is unlikely to be corrupted by a bad surface air temperature observation. I come back to this in the specific comments below

The other main comment is that aspects of the presentation should be improved, particularly many of the remaining diagrams. Nevertheless, the text is generally well written. I recommend this paper be accepted subject to significant/major revision.

*We thank the reviewer for the time taken to review our manuscript and their very useful feedback and helpful suggestions. Please see the responses and amendments below all made in this blue for ease of interpretation:*

Specific Comments

1. Lines 30- 62 of Introduction and/or Table 2. Somewhere about here it would be very useful to add the latitude, longitude and height location of Kilkenny Castle. In Table 2 it would add useful information to add these details for all the other stations shown – much as done in Table 3.

   *Both added with thanks. With regard to table 2- Lat, Long and height ASL has been added for those cases where the exact location is known. For other cases these fields are left blank.*

2. Figure 1. It is not so easy to see on a printed copy where Kilkenny Castle is located on either map. Perhaps an arrow and superimposed text of suitable size saying "Kilkenny Castle" could be added.

   *Both added with thanks and a zoom out now also used. With regard to table 2- Lat, Long and height ASL has been added for those cases where the exact location is known.*

3. Figure 2. The coloured dots would be easier to see if the green background was much lighter.

   *Amended with thanks. Revised figure is now much clearer and hopefully addresses this point satisfactorily.*

4. Figure 3. This diagram is not very clear on a printed copy, mainly because of the grey background. This can be fixed by making the graph lines thicker. The key is odd: the heading should be "stations" not "variable" and Kilkenny has a coloured line in the key, even though there are no data for this date, except the single point highlighted in the text. On the left, the label should add "degrees" before "Celsius". In the caption "air" should be added after "surface".

   *All amended with thanks. The revised figure is now much clearer and addresses all substantive points raised in this comment.*

5. Figure 4 and similar Figures. This is my most important comment. On the main Climate of the Twentieth Century Reanalysis v3 NOAA web site atmospheric temperature analyses are available in ensemble mean form for 8 times of the day for each day until 2015. I strongly recommend that 500hPa height be replaced by an

appropriate relatively low-level temperature (850hPa temperature or lower, as long as not significantly affected by station biases). This would be done (probably) for 1500 hours for each sub-diagram shown in Figure 4 and the similar figures. Ensemble mean PMSL would need to be be plotted for the same time of day for consistency. This would add genuine new information about the relative likelihood of the hottest temperature extremes. This type of information could become an important part of the evidence for daily surface air temperature extremes more generally. As far as I can see, appropriate sub-daily maps can be easily plotted for such data using the on line facility at https://www.psl.noaa.gov/data/20thC_Rean/ . After 2015, similar data should be available from ERA5.

*Amended with thanks. The following process was conducted in order to produce the new reanalysis-based plots:*

*This research paper used two reanalysis products,*
- *NOAA-CIRES-DOE Twentieth Century Reanalysis (V3)*
- *ECMWF ERA5 Reanalysis V5, ( ECMWF Reanalysis v5 (ERA5))*
- *The data plotted from the 1880s -1976 are NOAA*
- *The data plotted from 1983-2018 are ERA5*

*Note that the NOAA product does not extend beyond 2015 and therefore a switch to ERA5 was necessary to include post-2015 data. On this basis we chose to use ERA5 for any data post-1979 when the ERA5 fully documented product is available.*

*NOAA:*
- *The variables prmsl and 850 hpa tempertures were downloaded as Netcdf files.*
- *Files were imported into R and opened, processed and plotted.*
- *The variable prmsl was converted into mslp.*
- *The variable temperature was changed from kelvin to degrees celsius.*
- *The variables were then plotted on a map with temperature shown as colors and mslp as contours.*

*ERA5:*
- *The variables msl and 850 hpa tempertures were downloaded as Netcdf files.*
- *Files were imported into R and opened, processed and plotted.*
- *As ERA5 is already in msl no processing was needed for this variable.*
- *The variable temperature was changed from kelvin to degrees celsius.*
- *The variables were then plotted on a map with temperature shown as colors and mslp as contours.*

*These revised plots should better show what the reviewer requests*

6. Figure 5. On a printed copy, the red circle is less clear on some sub-diagrams than it should be. These are good diagrams otherwise. It might be useful to add combined simultaneous differences for all stations in a sixth panel -currently blank.
*Amended with thanks. The sixth panel has been added.*

7. Figure 7. Where relevant, I have the same comments about the graphs as for Figure 3.
*Amended with thanks.*

8. Figure 10. Same comments about the red dots as for Figure 5.

*Amended with thanks.*

Minor comments

1. At Line 241, Parker (1994) is missing from the Reference list.

*Added, thank you for catching that.*

**Reviewer 2**

This analysis reassess the 1887 high temperature record of Kilkenny, Ireland, using different datasets including the early-year records, current observations and reanalysis product, and concludes that this record and other early observations of extreme high temperature are not sufficiently reliable. The authors instead suggest the 1976 heat record at Boora as the highest national temperature measurement. The analysis is interesting, and the conclusion is also more or less convincing. This is not an easy work anyway. One of my concerns is the locality of the study, and thereby the narrowness nature of its significance if it is published in the CP. In addition, I find a few of other issues which demand further clarification.

*We thank reviewer 2 for the time taken to review our manuscript and the positive response provided. Please see the responses and amendments below all made in this blue for ease of interpretation:*

1. The Kilkenny observational site is located in the centre of Kilkenny Town. Was it also located in the town centre in 1887? What was the extent and population of the town in the early time? If the town has not changed much in urban area and population, would the central location of the observation in 1887 be more productive to the higher temperature records compared to other observational stations in the nearby cities and towns of similar size?

*The Kilkenny observational site is currently located in a private residence, which is believed to be a different location to the recording of 1887. However, the specific location of the 1887 record is not known within the Kilkenny Caste grounds.*

*In 1841, the population of Kilkenny City was approximately 19,071, decreasing to 15,257 in 1851 (McDill, 2021). In 1891, the population total of the county of Kilkenny was 87,261 (CSO, 2004). According to the 2016 census, Kilkenny City has a population of 25,512 (City Population, 2021). Therefore, it can be stated that the population has grown in Kilkenny City by over 10,000 since the mid-1800s. Since the 1800s, new buildings and developments have occurred in Kilkenny City. Although many historic buildings remain, ultimately the city is not the same as it was in 1887.*

*The closest weather station to Kilkenny Castle is Kilkenny Greenhills, which became a climatological station with daily observations from manual instruments being submitted to Met Eireann in 2010 (Kilkenny Weather, 2019). The maximum temperature recorded at this station since 2010, was 29.6 °C in 2013 and 2021. Other weather stations in Kilkenny do not record daily maximum temperatures. A figure has been added to illustrate the relative locations of the castle and the newer site and discussion around the potential similarities and distinctions and potential limitations has been added.*

2. The application of 20CR to show the synoptic situation seems not so persuasive, I feel. It indicates a favorable weather condition for the extreme hot and dry event, but when the authors said that the condition is not uncommon in the summer season over Ireland, they did not exhibit an example of the synoptic situation at present. Is there any similar or even more extreme high pressure system over the study region during the same length of periods in modern time?

*Thank you, maps have been replaced with MSLP and 850hPa temperature reanalysis. Further details on the new approach are given in the response to reviewer 1. The temperature extremes are principally a result of advection of airmasses rather than simply surface pressure. It is the particular circumstances of advection of airmasses from Africa / Southern Europe that can lead to heat extremes. The synoptic discussion aspects have been accordingly strengthened where we felt they were potentially causing this issue of interpretation.*

3. The exercise to compare the JJA season differences of maximum temperature between Kilkenny Greenshill and the four stations is convincing procedure. However, the representativeness/proximity of Kilkenny Greenshill to Kilkenny Castle would be important. The authors did not well document this. It would be good to at least give a large-scale map showing the specific locations of the two sites. Besides, all the data used should be evaluated for their quality and homogeneousness.

*Thanks for spotting this. We have added a map showing the proximity between these two stations as a new figure. We have also added a discussion of possible similarities / differences and caveats.*

4. Given Kilkenny Greenshill could well represent Kilkenny Castle, there are other issues that should be clarified. 1) Birr seems mostly close to Kilkenny Greenshill, with both being inland stations, but its modern data was not shown, why?

*Thank you, we have now provided information in the paper as to why Birr was excluded from the modern-day frequency distribution. The modern data was only available for the 2010 summer at Birr which was insufficient to analyse the frequency distributions. As a result, Birr station was excluded from the modern-day frequency distribution analysis. This detail has now been made clearer in the revised manuscript.*

2) For the two sites (Roches Point and Armagh) which is located inland or more closely to Kilkenny, the differences of Tmax between 1887 Kilkenny Castle record and those of the two stations are within 99% of the modern data, which means that the

differences are not impossible. If the reference sites are closer to Kilkenny, would the possibility for the 1887 Tmax differences to occur be higher? –

*The modern-day frequency distribution shown in Figure 5, does provide a possibility for the 1887 heat record of Kilkenny to have occurred however the frequency distribution also suggests that the 1887 heat record would be an extremely unlikely occurrence. The joint probability of exceeding that probability is obviously far more unlikely than each pairwise comparison. It is the combination of evidence from all the pairwise comparisons which, jointly, effectively rule out the recorded value as plausible. Furthermore, the June 1887 Tmax representation of the six selected weather stations shown in Figure 3, clearly has the Kilkenny heat record displayed as an outlier in comparison to its reference sites, If one or two sites being compared were at 99% of distribution that would be one thing, but for all comparator sites in all directions to be either at the limit of or entirely outside the expected distribution is quite another. It is both the extreme nature of individual pairwise comparisons, but more that in all directions these fall either outside or right in the extreme tail that raises serious questions around this possible record event.*

5. Finally, I would suggest that all the figures be improved for their quality and clarity.
   *Noted. Images have been improved and clarified according to the suggestion.*

---

## Author Response (AR2)

Dear Dr Dooley and co-authors

As you can see from the comments of the two expert reviewers, they find your manuscript well improved, but that it still lacks some information/analyses for it to be ready for publication. I realize that another major revision is not what you expected, but I'm confident that the suggested improvements will make your work even more relevant for a wider public. I agree with both reviewers that you should further attempt to set your study in a wider context, and I also suggest that you carefully consider the recommendations of reviewer #1 regarding the additional analyses (850 hPa temperature maps). While it may require a bit of extra work, it will give more weight to your results.

Good luck with the revision

Hans Linderholm, Editor

*Thanks for your feedback and also for the extension granted owing to covid infection and illness of critical authors. Responses are given in italics and this blue text throughout. Please note that owing to a hardware failure we had to manually reinsert the prior resubmission from pdf into the template. This may have led to some slight differences in the format of the paper but should have had no effect upon the contents.*

**Report #1**

*We thank the reviewer for the time taken to review our manuscript and their very useful feedback and helpful suggestions. We respond to specific points as they arise below and will of course attempt to improve the aspects of the presentation mentioned above (and below).*

Key Remarks

I have two main sets of remarks. Firstly, the authors have responded to the Editor and Reviewer 2 to place the paper better in a general context. However, I think they could go further. First, an improved title could reflect this change of emphasis. A suggested title, which might be improved, would be "Reassessing hottest temperature records using Ireland as an example." Secondly the Abstract could be modestly reconfigured, without making it significantly longer, by placing this work in the general context in the first few sentences rather than just at the end.

*We have modified the title (although it may now be too long) and moved up the final abstract sentence and reconfigured it to address this suggestion which we agree provides a better context to a generalist CP journal audience..*

My second comment would need more work. The authors have responded to my key comment and included 850hPa temperature maps. These data are clearly promising for aiding the arguments of this paper but are only used qualitatively. It is unclear whether the maps shown are one day averages or the 1500 GMT temperatures I recommended. This is not stated in the text and should be. 1500 GMT would correspond approximately to the time of maximum temperature in Ireland. It would take a little experimentation to see how much

difference these two alternative ways (daily average versus 1500GMT) of treating the 850hPa temperature data make.

*The maps are all for 1500 UTC (GMT) and this has been clarified accordingly in the text and figure captions.*

But the authors should make a more quantitative use of the 850hPa temperature data anyway. Quantitative use of 850hPa temperatures for a position above a given station could cast greater light on the relative likelihoods of the various high surface temperatures. One way is to adiabatically reduce the 850hPa temperatures to station heights (approximately when these are only known within a range of heights using local geography). These reduced values can then be plotted in one or more diagrams. The quasi-surface temperatures so created would only be approximate, but their relative values are likely to significantly aid judgement as to which which surface record to choose as being really the hottest.

*The issue here is one of a scale mismatch between the 20CRv3 product (available as a 1 degree resolution product) and the point nature and spatial density of the station networks being considered. To do this would require some approach to spatial interpolation to differentiate data at different locations which may or may not be a valid application to 20CRv3 and would compound the uncertainty associated with an assumption of adiabatic relaxation from 850 hPa (top of the boundary layer) down to the surface. In unpublished aspects of Ian Gillespie's PhD thesis under the supervision of co-author Peter Thorne on the use of 20CRv3 to homogenise surface temperatures the use of temperatures at greater heights than 2m was considered and shown to be a worse predictor of station level temperature timeseries than use of 20CRv3 2m temperatures. For these reasons while we see the methodological merit in the suggestion the available data at our disposal and our understanding of the relationship in 20CRV3 between surface temperatures at point locations and temperatures at various heights aloft suggest the uncertainty would be considerable.*

A slightly less good but simpler alternative would be to plot the set of 850hPa temperatures without reduction to a surface value and compare these. This may still reflect the relative levels of surface temperature adequately given the likely modest elevation differences between Irish stations. However, I am not sure about this. Such a methodology could be useful for studying hottest surface regional temperatures more generally, at least in the extra-tropics.

*We agree that 850hPa temperatures are informative and that is why we included them in the plots. For the two cases where the temporal evolution is considered we have added a set of maps at finer resolution as additional figures. We have also added discussion of 850hPa temperatures usefulness when introducing the first such map.*

Detailed remarks

1. Table 2 should include the approximate heights of each station where not known exactly or a perhaps a range of likely heights based on local geography.

*We have revisited and improved the location metadata in this table and its consistency*

2. Figure 2. Kilkenny station should be marked particularly clearly on this map, e.g. as a purple dot with a black ring around it like Figure 3. The Figure would be even clearer if a paler green was used.

*Figure 2 has been reedited to reflect these changes. The Kilkenny marker is now a red dot with a black marker around it, while the island colour is now a pale green to better denote the stations.*

3. Lines 185 onward. It should be made clear whether a daily average of the reanalyses is used or even better the 1500GMT (or nearest available) value as both reanalyses are available every three hours. Presumably the ensemble reanalysis mean is used – this should be clarified, mentioning ensemble size.

*Thank you. This research paper gathered data at 1500GMT using the ensemble mean. This has been clarified in both the text and the revised figure captions.*

4. Figure 4 and similar figures. The mapped variation of 850hPa temperature over Ireland would be even clearer if a somewhat reduced numerical scale range of colours (currently 0-25oC) was used. This will give even better differentiation of 850hPa temperature colours over Ireland. This might be helped for the map as a whole by reducing the geographical area covered to some extent. Again, arguably 1500 GMT (or similar) temperatures and PMSL would be better.

*We thank the reviewer for this feedback the larger geographical domain of interest was selected as we feel this provides a better synoptic picture of which air masses were most influential on the dates of interest. We have also added two figures with a zoom in to aid the reader to contextualise the 2 19th century candidates from Kilkenny Castle and Phoenix Park.*

From the authors comment in the body of text "Adiabatically reduce the 850hPa temperatures to station heights (approximately when these are only known within a range of heights using local geography":

*We thank you for your comment. See response above to the main comment to the same effect.*

5. About line 200. There needs to be an explicit discussion of what the maps of 850Pha temperatures show. This seems to be missing.

*Added with thanks.*

6. Figure 6. Needs labelling a-f for clarity. There is no mention in the caption of the last sub figure (nominally Fig.6f) or even a discussion of it in the text. But I think this sub figure should be kept.

*Figure has been amended as requested and reference has been given to 6f in the text.*

7. Line 262. "Recency" will not be understood by some non-native English speakers and should be replaced.

*Noted and so clarified.*

8. Line 272. The UK Met Office was founded in 1854 and so this statement is not true of the 1856 RCS Dublin record.

*Text has been edited to reflect this information.*

9. Lines 284-293. This discussion is not quite complete. Depending on geography, it is possible to have a very high temperature at a coastal station given a long a long upstream land passage of the air combined with an offshore wind. This should be clarified.

*Thank you, re-edited to include this.*

10. Table 3 and discussion. It would help to add a map of these stations when discussing the Boora record with the warmest record (currently Boora) shown more prominently as suggested for Fig 2.

*Added with thanks.*

Chris Folland 24 March 2022

**Report #2**

The broader implications of the work may be further explored in the context of global and regional warming, or of long-term extreme temperature change detection, in addition to the potential applicability of the method to surface air temperature records in other regions.

*We thank reviewer 2 for the time taken to review our revised manuscript and the positive response provided. We have tried to strengthen the final concluding paragraph to reflect their suggestion which is where we felt it best placed to add this valuable additional context.*

---

## Author Response (AR3)

*Dear Hans,*

*As per the editor's suggestions a note has been made on the new potential record achieved after the submission of this paper. The manuscript has also been updated to follow the referencing style required by the journal along with the other minor edits suggested.*

*With thanks,*

*Katherine*